# Efficiently Learning Fourier Sparse Set Functions

**Andisheh Amrollahi** *
ETH Zurich
Zurich, Switzerland
amrollaa@ethz.ch

**Amir Zandieh** *
EPFL
Lausanne, Switzerland
amir.zandieh@epfl.ch

**Michael Kapralov**[†]
EPFL
Lausanne, Switzerland
michael.kapralov@epfl.ch

**Andreas Krause**
ETH Zurich
Zurich, Switzerland
krausea@ethz.ch

## Abstract

Learning set functions is a key challenge arising in many domains, ranging from sketching graphs to black-box optimization with discrete parameters. In this paper we consider the problem of efficiently learning set functions that are defined over a ground set of size $n$ and that are sparse (say $k$-sparse) in the Fourier domain. This is a wide class, that includes graph and hypergraph cut functions, decision trees and more. Our central contribution is the first algorithm that allows learning functions whose Fourier support only contains low degree (say degree $d = o(n)$) polynomials using $O(kd \log n)$ sample complexity and runtime $O(kn \log^2 k \log n \log d)$. This implies that sparse graphs with $k$ edges can, for the first time, be learned from $O(k \log n)$ observations of cut values and in linear time in the number of vertices. Our algorithm can also efficiently learn (sums of) decision trees of small depth. The algorithm exploits techniques from the sparse Fourier transform literature and is easily implementable. Lastly, we also develop an efficient robust version of our algorithm and prove $\ell_2/\ell_2$ approximation guarantees without any statistical assumptions on the noise.

## 1 Introduction

How can we learn the structure of a graph by observing the values of a small number of cuts? Can we learn a decision tree efficiently by observing its evaluation on a few samples? Both of these important applications are instances of the more general problem of learning set functions.

Consider a set function which maps subsets of a ground set $V$ of size $n$ to real numbers, $x : 2^V \to \mathbb{R}$. Set functions that arise in applications often exhibit structure, which can be effectively captured in the Fourier (also called Walsh-Hadamard) basis. One common studied structure for set functions is *Fourier sparsity* [2]. A $k$-Fourier-sparse set function contains no more than $k$ nonzero Fourier coefficients. A natural example for $k$-Fourier-sparse set functions are cut functions of graphs with $k$ edges or evaluations of a decision tree of depth $d$ [7, 8, 12]. The cut function of a graph only contains polynomials of degree at most two in the Fourier basis and in the general case, the cut function of a hypergraph of degree $d$ only contains polynomials of degree at most $d$ in the Fourier basis [12]. Intuitively this means that these set functions can be written as sums of terms where each term depends on at most $d$ elements in the ground set. Also a decision tree of depth $d$ only contains polynomials of degree at most $d$ in the Fourier basis [7][8]. Learning such functions has recently

[†]Supported by ERC Starting Grant SUBLINEAR.

found applications in neural network hyper-parameter optimization [5]. Therefore, the family of Fourier sparse set functions whose Fourier support only contains low order terms is a natural and important class of functions to consider.

**Related work**   One approach for learning Fourier sparse functions uses Compressive Sensing (CS) methods [12]. Suppose we know that the Fourier transform of our function $\widehat{x}$ is $k$-sparse i.e. $|\mathrm{supp}(\widehat{x})| \leq k$, and $\mathrm{supp}(\widehat{x}) \subseteq P$ for some known set $P$ of size $p$. In [12] it is shown that recovery of $\widehat{x}$ is possible (with high probability) by observing the value of $x$ on $O(k \log^4 p)$ subsets chosen independently and uniformly at random. They utilize results from [10, 13] which prove that picking $O(k \log^4 p)$ rows of the Walsh-Hadamard matrix independently and uniformly at random results in a matrix satisfying the RIP which is required for recovery. For the case of graphs $p = \binom{n}{2} = O(n^2)$ and one can essentially learn the underlying graph with $O(k \log^4 n)$ samples. In fact this result can be further improved, and $O(k \log^2 k \log n)$ samples suffice [4]. Computationally, for the CS approach, one may use matching pursuit which takes $\Omega(kp)$ time and thus results in runtime of $\Omega(kn^d)$ for $k$ Fourier sparse functions of order $d$. This equals $\Omega(kn^2)$ for graphs, where $d = 2$. In [12], proximal methods are used to optimize the Lagrangian form of the $\ell_1$ norm minimization problem. Optimization is performed on $p$ variables which results in $\Omega(n^2)$ runtime for graphs and $\Omega(n^d)$ time for the general order $d$ sparse recovery case. Hence, these algorithms scale *exponentially* with $d$ and have *at least quadratic dependence on $n$* even in the simple case of learning graph cut functions.

There is another line of work on this problem in the sparse Fourier transform literature. [11] provides a non-robust version of the sparse Walsh Hadamard Transform (WHT). This algorithm makes restrictive assumptions on the signal, namely that the $k$ non-zero Fourier coefficients are chosen uniformly at random from the Fourier domain. This is a strong assumption that does not hold for the case of cut functions or decision trees. This work is extended in [4] to a robust sparse WHT called SPRIGHT. In addition to the the random uniform support assumption, [4] further presumes that the Fourier coefficients are finite valued and the noise is Gaussian. Furthermore, all existing sparse WHT algorithms are unable to exploit low-degree Fourier structure.

**Our results**   We build on techniques from the sparse Fourier transform literature [3, 6, 2] and develop an algorithm to compute the Walsh-Hadamard transform (WHT) of a $k$-Fourier-sparse signal whose Fourier support is constrained to low degree frequencies (low degree polynomials). For recovering frequencies with low degree, we utilize ideas that are related to compressive sensing over finite fields [1]. We show that if the frequencies present in the support of $\widehat{x}$ are of low order then there exists an algorithm that computes WHT in $O(kn \log^2 k \log n \log d)$ time using $O(kd \log n)$ samples. As opposed to [11], we avoid distributional assumptions on the support using hashing schemes. Our approach is the first one to achieve the sampling complexity of $O(kd \log n)$. Moreover its running time scales *linearly* in $n$ and there is no exponential dependence on $d$. For the important special case of graphs, where $d = 2$, our sampling complexity is near optimally $O(k \log n)$ and our runtime is $O(kn \log^2 k \log n)$ which is strictly better than CS methods which take at least quadratic time in $n$. This allows us to learn sparse graphs which have in the range of $800$ vertices in $\approx 2$ seconds whereas the previous methods [12] were constrained to the range of $100$ for similar runtimes.

For the case where $\widehat{x}$ is not exactly $k$-sparse, we provide novel robust algorithms that recover the $k$ dominant Fourier coefficients with provable $\ell_2/\ell_2$ approximation guarantees. We provide a robust algorithm using appropriate hashing schemes and a novel analysis. We further develop a robust recovery algorithm that uses $O(kd \log n \log(d \log n))$ samples and runs in time $O\left(nk \log^3 k + nk \log^2 k \log n \log(d \log n) \log d\right)$.

## 2   Problem Statement

Here we define the problem of learning set functions. Consider a set function which maps subsets of a ground set $V \triangleq \{1, \ldots, n\} = [n]$ of size $n$ to real numbers, $x : 2^V \to \mathbb{R}$. We assume oracle access to this function, that is, we can observe the function value $x(A)$ for any subset $A$ that we desire. The goal is to learn the function, that is to be able to evaluate it for all subsets $B \subseteq V$. A problem which has received considerable interest is learning *cut functions* of sparse (in terms of edges) graphs [12]. Given a weighted undirected graph $G = (V, E, w)$, the cut function associated to $G$ is defined as $x(A) = \sum_{s \in A, t \in V \setminus A} w(s, t)$, for every $A \subseteq V$.

Note that we can equivalently represent each subset $A \subseteq V$ by a vector $t \in \mathbb{F}_2^n$ which is the indicator of set $A$. Here $\mathbb{F}_2 = \{0, 1\}$ denotes the finite field with 2 elements. Hence the set function can be viewed as $x : \mathbb{F}_2^n \to \mathbb{R}$. We denote the Walsh-Hadamard transform of $x : \mathbb{F}_2^n \to \mathbb{R}$ by $\widehat{x} : \mathbb{F}_2^n \to \mathbb{R}$. It is defined as:

$$\widehat{x}_f = \frac{1}{\sqrt{N}} \sum_{t \in \mathbb{F}_2^n} x_t \cdot (-1)^{\langle f, t \rangle} \quad , f \in \mathbb{F}_2^n.$$

The inner product $\langle f, t \rangle$ throughout the paper is performed modulo 2.

The Fourier transform of the graph cut function $\widehat{x}$ is the following,

$$\widehat{x}_f = \begin{cases} \frac{1}{2} \sum_{s,t \in V} w(s,t) & \text{if } f = (0, \ldots, 0) \\ -w(s,t)/2 & \text{if } f_s = f_t = 1 \text{ and } f_i = 0 \ \forall i \neq s, t \ . \\ 0 & \text{otherwise} \end{cases}$$

It is clear that the Fourier support of the cut function for graph $G$ contains only $|E| + 1$ nonzero elements (and hence it is *sparse*). Furthermore, the nonzero Fourier coefficients correspond to frequencies with hamming weights at most 2.

One of the classes of set functions that we consider is that of **exactly low order Fourier sparse** functions. Under this model we address the following problem:

> **Input:** oracle access to $x : \mathbb{F}_2^n \to \mathbb{R}$
> such that $\|\widehat{x}\|_0 \leq k$ and $|f| \leq d$ for all $f \in \text{support}(\widehat{x})$ (1)
> **Output:** nonzero coefficients of $\widehat{x}$ and their corresponding frequencies

where $|f|$ denotes the *Hamming weight* of $f$.

We also consider the **robust** version of problem (1) where we only have access to noisy measurements of the input set function. We make no assumption about the noise, which can be chosen adversarially. Equivalently one can think of a general set function whose spectrum is well approximated by a low order sparse function which we refer to as *head*. *Head* of $\widehat{x}$ is just the top $k$ Fourier coefficients $\widehat{x}_f$ such that the frequency has low Hamming weight $|f| \leq d$. We refer to the noise spectrum as *tail*.

**Definition 1** (Head and Tail norm). For all integers $n$, $d$, and $k$ we define the *head* of $\widehat{x} : \mathbb{F}_2^n \to \mathbb{R}$ as,

$$\widehat{x}_{head} := \arg \min_{\substack{y : \mathbb{F}_2^n \to \mathbb{R} \\ \|y\|_0 \leq k \\ |j| \leq d \text{ for all } j \in \text{supp}(y)}} \|\widehat{x} - y\|_2.$$

The *tail norm* of $\widehat{x}$ is defined as, $\text{Err}(\widehat{x}, k, d) := \|\widehat{x} - \widehat{x}_{head}\|_2^2$.

Since the set function to be learned is only *approximately* in the low order Fourier sparse model, it makes sense to consider the *approximate* version of problem (1). We use the well known $\ell_2/\ell_2$ *approximation* to formally define the **robust** version of problem (1) as follows,

> **Input:** oracle access to $x : \mathbb{F}_2^n \to \mathbb{R}$
> **Output:** function $\widehat{\chi} : \mathbb{F}_2^n \to \mathbb{R}$ (2)
> such that $\|\widehat{\chi} - \widehat{x}\|_2^2 \leq (1 + \delta)\text{Err}(\widehat{x}, k, d)$,
> $|f| \leq d$ for all $f \in \text{support}(\widehat{\chi})$

Note that no assumptions are made about the function $x$ and it can be any general set function.

## 3 Algorithm and Analysis

In this section we present our algorithm and analysis. We use techniques from the sparse FFT literature [3, 6, 2]. Our main technical novelty is a new primitive for estimating a low order frequency, i.e., $|f| \leq d$, efficiently using an optimal number of samples $O(d \log n)$ given in Section 3.1. This primitive relies heavily on the fact that a low order frequency is constrained on a subset of size $\binom{n}{d}$ as opposed to the whole universe of size $2^n$. We show that problem (1) can be solved quickly and using a few samples from the function $x$ by proving the following theorem,

**Theorem 2.** *For any integers $n$, $k$, and $d$, the procedure* EXACTSHT *solves problem* (1) *with probability* $9/10$. *Moreover the runtime of this algorithm is* $O\left(kn \log^2 k \log n \log d\right)$ *and the sample complexity of this procedure is* $O\left(kd \log n\right)$.

We also show that problem (2) can be solved efficiently by proving the following theorem in the full version of this paper,

**Theorem 3.** *For any integers $n$, $k$, and $d$, the procedure* ROBUSTSHT *solves problem* (2) *with probability* $9/10$. *Moreover the runtime of this procedure is* $O\left(nk \log^3 k + nk \log^2 k \log n \log(d \log n) \log d\right)$ *and the sample complexity of the procedure is* $O\left(kd \log n \log(d \log n)\right)$.

**Remark:** This theorem proves that for any arbitrary input signal, we are able to achieve the $\ell_2/\ell_2$ guarantee using $O\left(kd \cdot \log n \cdot \log(d \log n)\right)$ samples. Using the techniques of [9] one can prove that the sample complexity is optimal up to $\log(d \log n)$ factor. Note that it is impossible to achieve this sample complexity without exploiting the low degree structure of the Fourier support.

## 3.1 Low order frequency recovery

In this section we provide a novel method for recovering a frequency $f \in \mathbb{F}_2^n$ with bounded Hamming weight $|f| \leq d$, from measurements $\langle m_i, f \rangle$ $i \in [s]$ for some $s = O(d \log n)$. The goal of this section is to design a measurement matrix $M \in \mathbb{F}_2^{s \times n}$ with small $s$, such that for any $f \in \mathbb{F}_2^n$ with $|f| \leq d$ the following system of constraints, with constant probability, has a unique solution $j = f$ and has an efficient solver,

$$j \in \mathbb{F}_2^n \text{ such that } \begin{cases} Mj = Mf \\ |j| \leq d \end{cases}.$$

To design an efficient solver for the above problem with optimal $s$, we first need an optimal algorithm for recovering frequencies with weight one $|f| \leq 1$. In this case, we can locate the index of the nonzero coordinate of $f$ optimally via binary search using $O(\log n)$ measurements and runtime.

**Definition 4** (Binary search vectors). For any integer $n$, the ensemble of vectors $\{v^l\}_{l=0}^{\lceil \log_2 n \rceil} \subseteq \mathbb{F}_2^n$ corresponding to binary search on $n$ elements is defined as follows. Let $v^0 = \{1\}^n$ (the all ones vector). For every $l \in \{1, \cdots, \lceil \log_2 n \rceil\}$ and every $j \in [n]$, $v_j^l = \left\lfloor \frac{(j \mod 2^l)}{2^{l-1}} \right\rfloor$.

**Lemma 5.** *There exists a set of measurements $\{m_i\}_{i=1}^s$ for $s = \lceil \log_2 n \rceil + 1$ together with an algorithm such that for every $f \in \mathbb{F}_2^n$ with $|f| \leq 1$ the algorithm can recover $f$ from the measurements $\langle f, m_i \rangle$ in time $O(\log_2 n)$.*

To recover a frequency $f$ with Hamming weight $d$, we hash the coordinates of $f$ randomly into $O(d)$ buckets. In expectation, a constant fraction of nonzero elements of $f$ get isolated in buckets, and hence the problem reduces to the weight one recovery. We know how to solve this using binary search as shown in Lemma 5 in time $O(\log n)$ and with sample complexity $O(\log n)$. We recover a constant fraction of the nonzero indices of $f$ and then we subtract those from $f$ and recurse on the residual. The pseudocode of the recovery procedure is presented in Algorithm 1.

**Lemma 6.** *For any integers $n$ and $d$, any power of two integer $D \geq 128d$, and any frequency $f \in \mathbb{F}_2^n$ with $|f| \leq d$, the procedure* RECOVERFREQUENCY *given in Algorithm 1 outputs $f$ with probability at least $7/8$, if we have access to the following,*

1. *For every $r = 0, 1, \cdots, \log_4 D$, a hash function $h_r : [n] \rightarrow [D/2^r]$ which is an instance from a pairwise independent hash family.*

2. *For every $l = 0, 1, \cdots, \lceil \log_2 n \rceil$ and every $r = 0, 1, \cdots, \log_4 D$, the measurements $\phi_r^l(i)$ that are equal to $\phi_r^l(i) = \sum_{j \in h_r^{-1}(i)} f_j \cdot v_j^l$ for every $i \in [D/2^r]$.*

*Moreover, the runtime of this procedure is $O(D \log D \log n)$ and the number of measurements is $O(D \log n)$.*

*Proof.* The proof is by induction on the iteration number $r = 0, 1, \cdots, T$. We denote by $\mathcal{E}_r$ the event $|f - \tilde{f}^{(r)}| \leq \frac{d}{4^r}$, that is the sparsity goes down by a factor of $4$ in every iteration up to $r^{th}$ iteration. The inductive hypothesis is $\Pr[\mathcal{E}_{r+1}|\mathcal{E}_r] \geq 1 - \frac{1}{16 \cdot 2^r}$.

---
**Algorithm 1** RECOVERFREQUENCY
---
**input**: power of two integer $D$, hash functions $h_r : [n] \to [D/2^r]$ for every $r \in \{0, 1, \cdots, \log_4 D\}$, measurement vectors $\phi_r^l \in \mathbb{F}_2^{D/2^r}$ for every $l = 0, 1, \cdots \lceil \log_2 n \rceil$ and every $r = 0, 1, \cdots, \log_4 D$.
**output**: recovered frequency $\tilde{f}$.

1:  $\{v^l\}_{l=0}^{\lceil \log_2 n \rceil} \leftarrow$ binary search vectors on $n$ elements (Definition 4), $T \leftarrow \log_4 D$, $\tilde{f}^{(0)} \leftarrow \{0\}^n$
2:  **for** $r = 0$ to $T$ **do**
3:     $w \leftarrow \{0\}^n$.
4:     **for** $i = 1$ to $D/2^r$ **do**
5:        **if** $\phi_r^0(i) - \sum_{j \in h_r^{-1}(i)} \tilde{f}_j^{(r)} \cdot v_j^0 = 1$ **then**
6:           $index \leftarrow \{0\}^{\lceil \log_2 n \rceil}$, a $\lceil \log_2 n \rceil$ bits pointer.
7:           **for** $l = 1$ to $\lceil \log_2 n \rceil$ **do**
8:              **if** $\phi_r^l(i) - \sum_{j \in h_r^{-1}(i)} \tilde{f}_j^{(r)} \cdot v_j^l = 1$ **then**
9:                 $[index]_l \leftarrow 1$, set $l^{th}$ bit of $index$ to 1.
10:          $w(index) \leftarrow 1$, set the coordinate of $w$ positioned at $index$ to 1.
11:    $\tilde{f}^{(r+1)} \leftarrow \tilde{f}^{(r)} + w$.
12: **return** $\tilde{f}^{(T+1)}$.
---

Conditioning on $\mathcal{E}_r$ we have that $|f - \tilde{f}^{(r)}| \leq \frac{d}{4^r}$. For every $i \in [D/2^r]$ and every $l \in \{0, 1, \cdots, \lceil \log_2 n \rceil\}$ it follows from the definition of $\phi_r^l$ that,

$$\phi_r^l(i) - \sum_{j \in h_r^{-1}(i)} \tilde{f}_j^{(r)} \cdot v_j^l = \sum_{j \in h_r^{-1}(i)} \left( f_j - \tilde{f}_j^{(r)} \right) \cdot v_j^l.$$

Let us denote by $S$ the support of vector $f - \tilde{f}^{(r)}$, namely let $S = \text{supp}\left( f - \tilde{f}^{(r)} \right)$.

From the pairwise independence of the hash function $h_r$ the following holds for every $a \in S$,

$$\Pr[h_r(a) \in h_r(S \setminus \{a\})] \leq 2^r \cdot \frac{|S|}{D} \leq 2^r \cdot \frac{1}{128 \cdot 4^r} \leq \frac{1}{128 \cdot 2^r}.$$

This shows that for every $a \in S$, with probability $1 - \frac{1}{128 \cdot 2^r}$, the bucket $h_r(a)$ contains no other element of $S$. Because the vector $f - \tilde{f}^{(r)}$ restricted to the elements in bucket $h_r^{-1}(h_r(a))$ has Hamming weight one, for every $a \in S$,

$$\Pr \left[ \left| \left( f - \tilde{f}^{(r-1)} \right)_{h_r^{-1}(h_r(a))} \right| = 1 \right] \geq 1 - \frac{1}{128 \cdot 2^r}.$$

If the above condition holds, then it is possible to find the index of the nonzero element via binary search as in Lemma 5. The for loop in line 7 of Algorithm 1 implements this. Therefore with probability $1 - \frac{1}{16 \cdot 2^r}$ by Markov's inequality a $1 - 1/8$ fraction of the support elements, $S$, gets recovered correctly and at most $1/8$ fraction of elements remain unrecovered and possibly result in false positive. Since the algorithm recovers at most one element per bucket, the total number of falsely recovered indices is no more than the number of non-isolated buckets which is at most $1/8 \cdot |S|$. Therefore with probability $1 - \frac{1}{16 \cdot 2^r}$, the residual at the end of $r$th iteration has sparsity $1/8 \cdot |S| + 1/8 \cdot |S| = 1/4 \cdot |S|$, i.e. $\left| f - \tilde{f}^{(r+1)} \right| \leq \frac{|S|}{4} \leq \frac{d}{4^{r+1}}$. This proves the inductive step.

It follows from the event $\mathcal{E}_T$ for $T = \log_4 D$ that $\tilde{f}^{(T)} = f$, where $\tilde{f}^{(T)}$ is the output of Algorithm 1. The inductive hypothesis along with union bound implies that $\Pr\left[\bar{\mathcal{E}}_T\right] \leq \sum_{r=1}^{T} \Pr\left[\bar{\mathcal{E}}_r | \mathcal{E}_{r-1}\right] + \Pr\left[\bar{\mathcal{E}}_0\right] \leq \sum_{r=0}^{T} \frac{1}{16 \cdot 2^r} \leq 1/8$.

**Runtime:** the algorithm has three nested loops and the total number of repetitions of all loops together is $O(D \log n)$. The recovered frequency $\tilde{f}^{(r)}$ always has at most $O(D)$ nonzero entries therefore the time to calculate $\sum_{j \in h_r^{-1}(i)} \tilde{f}_j^{(r-1)} \cdot v_j^l$ for a fixed $r$ and a fixed $l$ and all $i \in [D/2^r]$ is $O(D)$. Therefore the total runtime is $O(D \log D \log n)$.

**Number of measurements:** the number of measurements is the total size of the measurement vectors $\phi_r^l$ which is $O(D \log n)$. $\qquad\square$

## 3.2 Signal reduction

We now develop the main tool for estimating the frequencies of a sparse signal, namely the HASH2BINS primitive. If we hash the frequencies of a $k$-sparse signal into $O(k)$ buckets, we expect most buckets to contain at most one of the elements of the support of our signal. The next definition shows how we compute the hashing of a signal in the time domain.

**Definition 7.** For every $n, b \in \mathbb{N}$, every $a \in \mathbb{F}_2^n$, and every $\sigma \in \mathbb{F}_2^{n \times b}$ and every $x : \mathbb{F}_2^n \to \mathbb{R}$, we define the hashing of $\widehat{x}$ as $u_\sigma^a : \mathbb{F}_2^b \to \mathbb{R}$, where $u_\sigma^a(t) = \sqrt{\frac{2^n}{2^b}} \cdot x_{\sigma t + a}$, for every $t \in \mathbb{F}_2^b$.

We denote by $B \triangleq 2^b$ the number of buckets of the hash function. In the next claim we show that the Fourier transform of $u_\sigma^a$ corresponds to hashing $\widehat{x}$ into $B$ buckets.

**Claim 8.** For every $j \in \mathbb{F}_2^b$, $\widehat{u}_\sigma^a(j) = \sum_{f \in \mathbb{F}_2^n : \sigma^\top f = j} \widehat{x}_f \cdot (-1)^{\langle a, f \rangle}$.

Let $h(f) \triangleq \sigma^\top f$. For every $j \in \mathbb{F}_2^b$, $\widehat{u}_\sigma^a$ is the sum of $\widehat{x}_f \cdot (-1)^{\langle a, f \rangle}$ for all frequencies $f \in \mathbb{F}_2^n$ such that $h(f) = j$, hence $h(f)$ can be thought of as the bucket that $f$ is hashed into. If the matrix $\sigma$ is chosen uniformly at random then the hash function $h(\cdot)$ is pairwise independent.

**Claim 9.** For any $n, b \in \mathbb{N}$, if the hash function $h : \mathbb{F}_2^n \to \mathbb{F}_2^b$ is defined as $h(\cdot) = \sigma^\top(\cdot)$, where $\sigma \in \mathbb{F}_2^{n \times b}$ is a random matrix whose entries are distributed independently and uniformly at random on $\mathbb{F}_2$, then for any $f \neq f' \in \mathbb{F}_2^n$ it holds that $\Pr[h(f) = h(f')] = \frac{1}{B}$, where the probability is over picking $n \cdot b$ random bits of $\sigma$.

---

**Algorithm 2** HASH2BINS

**input**: signal $x \in \mathbb{R}^{2^n}$, signal $\widehat{\chi} \in \mathbb{R}^{2^n}$, integer $b$, binary matrix $\sigma \in \mathbb{F}_2^{n \times b}$, shift vector $a \in \mathbb{F}_2^n$.
**output**: hashed signal $\widehat{u}_\sigma^a$.

1: Compute $\widehat{u}_\sigma^a = \mathsf{FHT}\left(\sqrt{\frac{2^n}{2^b}} \cdot x_{\sigma(\cdot)+a}\right)$.  $\qquad\qquad \triangleright$ FHT is the fast Hadamard transform algorithm
2: $\widehat{u}_\sigma^a(j) \leftarrow \widehat{u}_\sigma^a(j) - \sum_{f \in \mathbb{F}_2^n : \sigma^\top f = j} \widehat{\chi}_f \cdot (-1)^{\langle a, f \rangle}$ for every $j \in \mathbb{F}_2^b$.
3: **return** $\widehat{u}_\sigma^a$.

---

The HASH2BINS primitive computes the Fourier coefficients of the residue signal that are hashed to each of the buckets. We denote by $\widehat{\chi}$ the estimate of $\widehat{x}$ in each iteration. As we will see in Section 3.3, the recovery algorithm is iterative in the sense that we iterate over $\widehat{x} - \widehat{\chi}$ (the residue) whose sparsity is guaranteed to decrease by a constant factor in each step.

**Claim 10.** For any signal $x, \widehat{\chi} : \mathbb{F}_2^n \to \mathbb{R}$, integer $b$, matrix $\sigma \in \mathbb{F}_2^{n \times b}$, and vector $a \in \mathbb{F}_2^n$ the procedure HASH2BINS$(x, \widehat{\chi}, b, \sigma, a)$ given in Algorithm 2 computes the following using $O(B)$ samples from $x$ in time $O(Bn \log B + \|\widehat{\chi}\|_0 \cdot n \log B)$

$$\widehat{u}_\sigma^a(j) = \sum_{f \in \mathbb{F}_2^n : \sigma^\top f = j} \widehat{(x - \chi)}_f \cdot (-1)^{\langle a, f \rangle}.$$

## 3.3 Exact Fourier recovery

In this section, we present our algorithm for solving the exact low order Fourier sparse problem defined in (1) and prove Theorem 2. Let $S \triangleq \mathrm{supp}(\widehat{x})$. Problem (1) assumes that $|S| \leq k$ and also for every $f \in S, |f| \leq d$. The recovery algorithm hashes the frequencies into $B = 2^b$ buckets using Algorithm 2. Every frequency in the support $f \in S$ is recoverable, with constant probability, if no other frequency from the support collides with it in the hashed signal. The collision event is formally defined below,

**Definition 11** (Collision). For any frequency $f \in \mathbb{F}_2^n$ and every sparse signal $\widehat{x}$ with support $S = \mathrm{supp}(\widehat{x})$, the collision event $E_{coll}(f)$ corresponding to the hash function $h(f) = \sigma^\top f$ holds iff $h(f) \in h(S \setminus \{f\})$.

**Claim 12** (Probability of collision). *For every $f \in \mathbb{F}_2^n$, if the hash function $h : \mathbb{F}_2^n \to \mathbb{F}_2^b$ is defined as $h(\cdot) = \sigma^\top(\cdot)$, where $\sigma \in \mathbb{F}_2^{n \times b}$ is a random matrix whose entries are distributed independently and uniformly at random on $\mathbb{F}_2$ then $\Pr[E_{coll}(f)] \le \frac{k}{B}$ (see Definition 11). The probability is over the randomness of matrix $\sigma$.*

If the hash function $h(\cdot) = \sigma^\top(\cdot)$ is such that the collision event $E_{coll}(f)$ does not occur for a frequency $f$, then it follows from Claim 8 and Definition 11 that for every $a \in \mathbb{F}_2^n$,

$$\widehat{u}_\sigma^a(h(f)) = \widehat{x}_f \cdot (-1)^{\langle a, f \rangle}.$$

Therefore, under this condition, the problem reduces to $d$-sparse recovery. If $a = \{0\}^n$ then, $\widehat{u}_\sigma^a(h(f)) = \widehat{x}_f$. Hence for any $m \in \mathbb{F}_2^n$, one can learn the inner product $\langle m, f \rangle$ by comparing the sign of $\widehat{u}_\sigma^m(h(f)) = \widehat{x}_f \cdot (-1)^{\langle m, f \rangle}$ and $\widehat{u}_\sigma^a(h(f))$. If the signs are the same then $(-1)^{\langle m, f \rangle} = 1$ meaning that $\langle m, f \rangle = 0$ and if the signs are different then $\langle m, f \rangle = 1$. In Section 3.1 we gave an algorithm for learning a low order frequency $|f| \le d$ from measurements of the form $\langle m, f \rangle$. So putting these together gives the inner subroutine for our sparse fast Hadamard transform, which performs one round of hashing, presented in Algorithm 3.

---

**Algorithm 3** SHTINNER

---

**input**: signal $x \in \mathbb{R}^{2^n}$, signal $\widehat{\chi} \in \mathbb{R}^{2^n}$, failure probability $p$, integer $b$, integer $d$.
**output**: recovered signal $\widehat{\chi}'$.

1: Let $\{v^l\}_{l=0}^{\lceil \log_2 n \rceil}$ be binary search vectors on $n$ elements (Definition 4).
2: $D \leftarrow$ smallest power of two integer s.t. $D \ge 128d$, $R \leftarrow \lceil 2 \log_2(1/p) \rceil$.
3: For every $r \in \{0, 1, \cdots, \log_4 D\}$ and every $s \in [R]$, let $h_r^s : [n] \to [D/2^r]$ be an independent copy of a pairwise independent hash function.
4: For every $r \in \{0, 1, \cdots, \log_4 D\}$, every $s \in [R]$, and every $j \in [D/2^r]$ let $w_{r,s}^j \in \mathbb{F}_2^n$ be the binary indicator vector of the set $h_r^s(j)^{-1}$.
5: For every $s \in [R]$, every $r \in \{0, 1, \cdots, \log_4 D\}$ and every $l \in \{0, 1, \cdots, \lceil \log_2 n \rceil\}$ and every $j \in [D/2^r]$, add $w_{r,s}^j \cdot v^l$ to set $A_s$.
6: Let $\sigma \in \mathbb{F}_2^{n \times b}$ be a random matrix. Each entry is independent and uniform on $\mathbb{F}_2$.
7: For every $a \in \cup_{s \in [R]} A_s$ compute $\widehat{u}_\sigma^a = $ HASH2BINS$(x, \widehat{\chi}, b, \sigma, a)$.
8: **for** $j = 1$ to $B$ **do**
9:      Let $L$ be an empty multi-set.
10:      **for** $s \in [R]$ **do**
11:          **for** every $r \in \{0, \cdots, \log_4 D\}$, every $i \in [D/2^r]$, and every $l \in \{0, \cdots, \lceil \log_2 n \rceil\}$ **do**
12:              **if** $\widehat{u}_\sigma^c(j) \ne 0$, where $c = \{0\}^n$ **then**
13:                  **if** $\widehat{u}_\sigma^c(j)$ and $\widehat{u}_\sigma^{w_{r,s}^i \cdot v^l}(j)$ have same sign **then** $\phi_r^l(i) \leftarrow 0$. **else** $\phi_r^l(i) \leftarrow 1$.
14:          $\tilde{f} \leftarrow$ RECOVERFREQUENCY $\left( D, \{h_r^s\}_{r=0}^{\log_2 D}, \left\{\{\phi_r^l\}_{r=0}^{\log_4 D}\right\}_{l=0}^{\lceil \log_2 n \rceil} \right)$.
15:          Append $\tilde{f}$ to multi-set $L$.
16:      $f \leftarrow \text{majority}(L)$
17:      $\widehat{\chi}'_f \leftarrow \widehat{u}^c(j)$, where $c = \{0\}^n$.
18: **return** $\widehat{\chi}'$.

---

**Lemma 13.** *For all integers $b$ and $d$, every signals $x, \widehat{\chi} \in \mathbb{R}^{2^n}$ such that $|\xi| \le d$ for every $\xi \in supp(\widehat{x - \chi})$, and any parameter $p > 0$, Algorithm 3 outputs a signal $\widehat{\chi}' \in \mathbb{R}^{2^n}$ such that $|supp(\widehat{\chi}')| \le |supp(\widehat{x - \chi})|$ and also for every frequency $f \in supp(\widehat{x - \chi})$, if the collision event $E_{coll}(f)$ does not happen then,*

$$\Pr\left[ \widehat{\chi}'_f = (\widehat{x - \chi})_f \right] \ge 1 - p.$$

*Moreover the sample complexity of this procedure is $O(Bd \log n \log \frac{1}{p})$ and also its time complexity is $O\left( B \log B(n + d \log n \log \frac{1}{p}) + nB \log n \log d \log \frac{1}{p} + \|\widehat{\chi}\|_0 \cdot n(\log B + \log n \log d \log \frac{1}{p}) \right)$.*

**Lemma 14.** *For any parameter $p > 0$, all integers $k$, $d$, and $b \ge \log_2(k/p)$, every signal $x, \widehat{\chi} \in \mathbb{R}^{2^n}$ such that $\|\widehat{x - \chi}\|_0 \le k$ and $|\xi| \le d$ for every $\xi \in supp(\widehat{x - \chi})$, the output of*

SHTINNER$(x, \widehat{\chi}, p, b, d)$, $\widehat{\chi}'$ satisfies the following with probability at least $1 - 32p$,

$$\|\widehat{x} - \widehat{\chi} - \widehat{\chi}'\|_0 \leq k/8.$$

Our sparse Hadamard transform algorithm iteratively calls the primitive SHTINNER to reduce the sparsity of the residual signal by a constant factor in every iteration. Hence, it terminates in $O(\log k)$ iterations. See Algorithm 4.

---

**Algorithm 4** EXACTSHT

---

**input**: signal $x \in \mathbb{R}^{2^n}$, failure probability $q$, sparsity $k$, integer $d$.
**output**: estimate $\widehat{\chi} \in \mathbb{R}^{2^n}$.
1: $p^{(1)} \leftarrow q/32, b^{(1)} \leftarrow \lceil \log_2 \frac{64k}{q} \rceil, w^{(0)} \leftarrow \{0\}^{2^n}, T \leftarrow \lceil \log_8 k \rceil$.
2: **for** $r = 1$ to $T$ **do**
3: $\quad \widetilde{\chi} \leftarrow$ SHTINNER$(x, w^{(r-1)}, p^{(r)}, b^{(r)}, d)$
4: $\quad w^{(r)} \leftarrow w^{(r-1)} + \widetilde{\chi}$.
5: $\quad p^{(r+1)} \leftarrow p^{(r)}/2, b^{(r+1)} \leftarrow b^{(r)} - 2$.
6: $\widehat{\chi} \leftarrow w^{(T)}$.
7: **return** $\widehat{\chi}$.

---

**Proof of Theorem 2:** The proof is by induction. We denote by $\mathcal{E}_r$ the event corresponding to $\|\widehat{x} - w^{(r)}\|_0 \leq \frac{k}{8^r}$. The inductive hypothesis is $\Pr[\mathcal{E}_r | \mathcal{E}_{r-1}] \geq 1 - 16p^{(r)}$. Conditioned on $\mathcal{E}_{r-1}$ we have that $\|\widehat{x} - w^{(r-1)}\|_0 \leq \frac{k}{8^{r-1}}$. The number of buckets in iteration $r$ of the algorithm is $B^{(r)} = 2^{b^r} \geq \frac{64k}{4^{r-1} \cdot q}$. Hence, it follows from Lemma 14, that with probability $1 - 32p^{(r)}$, $\|\widehat{x} - w^{(r)}\|_0 \leq \frac{k}{8^r}$. This proves the inductive step.

**Runtime and Sample complexity:** In iteration $r \in [\lceil \log_8 k \rceil]$, the size of the bucket $B^{(r)} = 2^{b^{(r)}} = \frac{64k}{q \cdot 4^r}$ and the error probability $p^{(r)} = \frac{q}{32 \cdot 2^r}$. Moreover at most $\sum_r B^{(r)}$ elements are added to $\widehat{\chi}$, hence we can assume that $\|\widehat{\chi}\|_0 \leq \frac{128k}{q}$. From Lemma 13 it follows that the total runtime is $O\left(kn \log^2 k \log n \log d\right)$.

The sample complexity of iteration $r$ is $O\left(\frac{kd}{2^r} \log n \log 2^r\right)$ hence the total sample complexity is dominated by the sample complexity of the first iteration which is equal to $O\left(kd \log n\right)$. $\quad \square$

## 4 Experiments

We test our EXACTSHT algorithm for graph sketching on a real world data set. We utilize the autonomous systems dataset from the SNAP data collection.[3] In order to compare our methods with [12] we reproduce their experimental setup. The dataset consists of 9 snapshots of an autonomous system in Oregon on 9 different dates. The goal is detect which edges are added and removed when comparing the system on two different dates. As a pre-processing step, we find the common vertices that exist on all dates and look at the induced subgraphs on these vertices. We take the symmetric differences (over the edges) of dates 7 and 9. Results for other date combinations can be found in the supplementary material. This results in a sparse graph (sparse in the number of edges). Recall that the running time of our algorithm is $O(kn \log^2 k \log n \log d)$ which reduces to $O(nk \log^2 k \log n)$ for the case of cut functions where $d = 2$.

### 4.1 Sample and time complexities as number of vertices varies

In the first experiment depicted in Figure 1b we order the vertices of the graph by their degree and look at the induced subgraph on the $n$ largest vertices in terms of degree where $n$ varies. For each $n$ we pick $e = 50$ edges uniformly at random. The goal is to learn the underlying graph by observing the values of cuts. We choose parameters of our algorithm such that the probability of success is at least 0.9. The parameters tuned in our algorithm to reach this error probability are the initial number

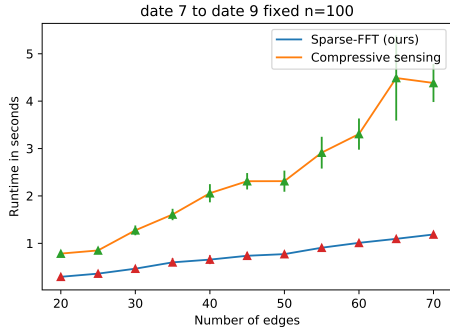

(a) Avg. time vs. no. edges

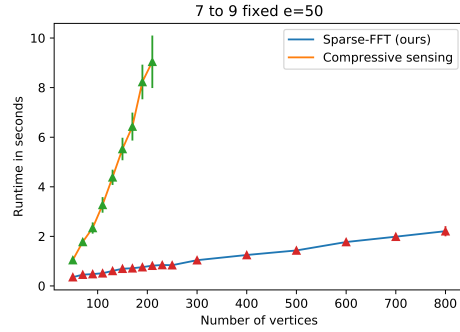

(b) Avg. time vs. no. vertices

Table 1: Sampling and computational complexity

| No. of vertices | CS method | | Our method | |
|---|---|---|---|---|
| | Runtime | Samples | Runtime | Samples |
| 70 | 1.14 | 767 | 0.85 | 6428 |
| 90 | 1.88 | 812 | 0.92 | 6490 |
| 110 | 3.00 | 850 | 0.82 | 6491 |
| 130 | 4.31 | 880 | 1.01 | 7549 |
| 150 | 5.34 | 905 | 1.16 | 7942 |
| 170 | 6.13 | 927 | 1.22 | 7942 |
| 190 | 7.36 | 947 | 1.18 | 7271 |
| 210 | 8.24 | 965 | 1.28 | 7271 |
| 230 | $*$ | $*$ | 1.38 | 7942 |
| 250 | $*$ | $*$ | 1.38 | 7271 |
| 300 | $*$ | $*$ | 1.66 | 8051 |
| 400 | $*$ | $*$ | 2.06 | 8794 |
| 500 | $*$ | $*$ | 2.42 | 8794 |
| 600 | $*$ | $*$ | 3.10 | 9646 |
| 700 | $*$ | $*$ | 3.35 | 9646 |
| 800 | $*$ | $*$ | 3.60 | 9646 |

of buckets the frequencies are hashed to and the ratio at which they reduce in each iteration. We plot running times as $n$ varies. We compare our algorithm with that of [12] which utilizes a CS approach. We fine-tune their algorithm by tuning the sampling complexity. Both algorithms are run in a way such that each sample (each observation of a cut value) takes the same time. As one can see our algorithm scales *linearly* with $n$ (up to log factors) whereas the CS approach scales *quadratically*. Our algorithm continues to work in a reasonable amount of time for vertex sizes as much as 900 in under 2 seconds. Error bars depict standard deviations.

In Table 1 we include both sampling complexities (number of observed cuts) and running times as $n$ varies. Our sampling complexity is equal to $O(k \log n)$. In practice they perform around a constant factor of 10 worse than the compressive sensing method, which are not provably optimal (see Section 1) but perform well in practice. In terms of computational cost, however, the CS approach quickly becomes intractable, taking large amounts of time on instance sizes around 200 and larger [12]. Asterisks in Table 1 refer to experiments that have taken too long to be feasible to run.

## 4.2 Time complexities as number of edges varies

Here we fix the number of vertices to $n = 100$ and consider the induced subgraph on these vertices. We randomly pick $e$ edges to include in the graph. We plot computational complexities. Our running time provably scales linearly in the number of edges as can be seen in Figure 1a.

## Footnotes

*The first two authors contributed equally

[3]snap.stanford.edu/data/

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
