[Supplementary Material]

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

*Proof.* Let us denote by $\tilde{f}$ what the algorithm recovers from the measurements. Let the measurements correspond to the binary search vectors $v^l$ for $l = 0, 1, \cdots \lceil \log_2 n \rceil$ (Definition 4). Note that for any $f$ with $|f| \leq 1$ and any vector $m \in \mathbb{F}_2^n$, if $\langle f, m \rangle = 1$ then $\text{supp}(f) \subseteq \text{supp}(m)$ and otherwise $\text{supp}(f) \subseteq \text{supp}(\bar{m})$ where $\bar{m}$ is the entrywise complement of $m$. Therefore the following algorithm can recover $f$,

1:   $\tilde{f} \leftarrow \{0\}^n$.
2:   **if** $\langle f, v^0 \rangle = 1$ **then**
3:       $index \leftarrow \{0\}^{\lceil \log_2 n \rceil}$, a $\lceil \log_2 n \rceil$ bits number initialized at 0.
4:       For every $l = 1, 2, \cdots, \lceil \log_2 n \rceil$, if $\langle f, v^l \rangle = 1$ then $[index]_l \leftarrow 1$, $l^{th}$ bit of $index$ sets to 1.
5:       $\tilde{f}(index) \leftarrow 1$, the nonzero coordinate of $\tilde{f}$ is positioned at $index$.

The proof of why the above algorithm works is by induction and fairly straightforward. Above algorithm runs in time $O(\log n)$. □

To recover a frequency $f$ with Hamming weight $d$, we hash the coordinates of $f$ randomly into $O(d)$ buckets. In expectation a constant fraction of nonzero elements of $f$ get isolated in buckets, and hence the problem reduces to the weight one recovery. We know how to solve this using binary search as shown in Lemma 5 in time $O(\log n)$ and with sample complexity $O(\log n)$. We recover a constant fraction of the nonzero indices of $f$ and then we subtract those from $f$ and recurse on the residual. We decrease the number of buckets we hash the coordinates into by a factor of $\frac{1}{2}$ in each step. We expect the sparsity of the frequency (the hamming weight of the frequency) to go down by a factor of $\frac{1}{4}$ in each step. The recovery procedure is presented in Algorithm 1.

**Lemma 6.** *For any integers $n$ and $d$ , any power of two integer $D \geq 128d$, and any frequency $f \in \mathbb{F}_2^n$ with $|f| \leq d$, the procedure RECOVERFREQUENCY given in Algorithm 1 outputs $f$ with probability at least $7/8$, if the following holds,*

1. *For every $r = 0, 1, \cdots, \log_4 D$, the hash function $h_r : [n] \to [D/2^r]$

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

*Proof.* We know that $x_t = \frac{1}{\sqrt{N}} \sum_{f \in \mathbb{F}_2^n} \widehat{x}_f \cdot (-1)^{\langle f, t \rangle}$. Hence it follows that:

$$x_{\sigma t + a} = \frac{1}{\sqrt{N}} \sum_{f \in \mathbb{F}_2^n} \widehat{x}_f \cdot (-1)^{\langle f, \sigma t + a \rangle} = \frac{1}{\sqrt{N}} \sum_{f \in \mathbb{F}_2^n} \widehat{x}_f \cdot (-1)^{\langle \sigma^\top f, t \rangle + \langle f, a \rangle} \tag{3}$$

By definition of the WHT we have that:

$$\widehat{u}_\sigma^a(j) = \frac{1}{\sqrt{B}} \sqrt{\frac{2^n}{2^b}} \sum_{t \in \mathbb{F}_2^b} x_{\sigma t + a} \cdot (-1)^{\langle j, t \rangle} = \frac{1}{B} \sum_{t \in \mathbb{F}_2^b} x_{\sigma t + a} \cdot (-1)^{\langle j, t \rangle} \tag{4}$$

Inserting Equation (3) into Equation (4) yields:

$$\widehat{u}_\sigma^a(j) = \frac{1}{B} \sum_{f \in \mathbb{F}_2^n} \widehat{x}_f (-1)^{\langle a, f \rangle} \sum_{t \in \mathbb{F}_2^b} (-1)^{\langle \sigma^\top f + j, t \rangle}$$

The second summation is zero $\sigma^\top f \neq j$ and equal to $2^b = B$ otherwise. Hence:

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

*Proof.* Because of the if condition in line 13 of the algorithm the empty buckets don't contribute to nonzero elements in the recovered signal $\widehat{\chi}'$. Hence, $|supp(\widehat{\chi}')| \leq |supp(\widehat{x - \chi})|$. Assume that $E_{coll}(f)$ does not hold for some arbitrary $f \in supp(\widehat{x - \chi})$. Therefore, for every $a \in \mathbb{F}_2^n$,

$$\widehat{u}_\sigma^a(h(f)) = (\widehat{x - \chi})_f \cdot (-1)^{\langle a, f \rangle}.$$

Fix one $s \in [R]$. For every $r \in \{0, 1, \cdots, \log_4 D\}$, every $i \in [D/2^r]$ and every $l \in \{0, 1, \cdots, \lceil \log_2 n \rceil\}$, the algorithm compares the sign of $\widehat{u}_\sigma^c(j)$ and $\widehat{u}_\sigma^{w_{r,s}^i \cdot v^l}(j)$, where $c = \{0\}^n$. If they have the same

sign then it means that $\langle f, w_{r,s}^i \cdot v^l \rangle = 0$ otherwise $\langle f, w_r^i \cdot v^l \rangle = 1$. Hence, $\phi_r^l(i) = \langle f, w_r^i \cdot v^l \rangle = \sum_{j \in h_r^{-1}(i)} f_j \cdot v_j^l$. Lemma 6 implies that $\textsc{RecoverFrequency}\left(D, \{h_r\}_{r=0}^{\log_2 D}, \left\{\{\phi_r^l\}_{r=0}^{\log_4 D}\right\}_{l=0}^{\lceil \log_2 n \rceil}\right)$ recovers $f$ with probability $1 - 1/8$ in each iteration of the for loop over $s \in [R]$. The algorithm repeats this independently for every $s \in [R]$ and then takes a majority vote over all the outputted frequencies by $\textsc{RecoverFrequency}$. Frequency $f$ in line 20 of the algorithm is the frequency which appears the most in the output of $\textsc{RecoverFrequency}$. The probability of failing to recover $f$ is the following,

$$\binom{R}{R/2} \cdot (1/8)^{R/2} \leq (1/2)^{R/2} \leq p.$$

Then the algorithm estimates the value of $\widehat{\chi}'_f$ in line 21 as $\widehat{\chi}'_f = \widehat{u}^c(h(f)) = \widehat{(x-\chi)}_f$ which is correct with probability one.

**Runtime:** Computing the hashing $\widehat{u}_\sigma^a$ with all the different shift parameters $a \in \cup_{s \in [R]} A_s$ is one of the most expensive operations in this procedure. For a fixed $a \in \mathbb{F}_2^n$, in order to compute $x_{\sigma t + a}$ we need $B$ time samples, one for each $t \in \mathbb{F}_2^b$. The computation of the indices $\sigma t$ and all $t \in \mathbb{F}_2^b$ is upper bounded by $O(nB \log B)$ operations. Given that we have computed $\sigma t$ for all $t \in \mathbb{F}_2^b$ and stored it in memory, for a fixed $a$ computing $\sigma t + a$ for all $t$ takes $O(B\|a\|_0)$ operation. Note that vectors $a \in \cup_{s \in [R]} A_s$ are sparse because for every $r, s, j$ the vector $w_{r,s}^j$ in line 5 of Algorithm 3 has only $2^r n/D$ non-zero entries. Therefore, the total complexity of forming the reduced signals $x_{\sigma t + a}$ for all $a \in \cup_{s \in [R]} A_s$ is $O(nB \log B + nB \log n \log d \log(1/p))$. The computational complexity of a fast Walsh Hadamard transform on $x_{\sigma t + a}$ is equal to $B \log_2 B$. Hence, the computational complexity of computing the hashings $\widehat{u}_\sigma^a$ is $O(B \log B(d \log n \log(1/p) + n) + nB \log n \log d \log(1/p))$.

We also need to subtract off the current estimate $\chi$. For each frequency $f \in \text{supp}(\widehat{\chi})$ we compute $j = \sigma^\top f$. This takes $O(\|\widehat{\chi}\|_0 n \log B)$ time in total. Next for each frequency $f \in \text{supp}(\widehat{\chi})$ and each $a \in \cup_{s \in [R]} A_s$ the inner product $\langle f, a \rangle$ needs to be computed. This takes total time of $O(\|\widehat{\chi}\|_0 \cdot n \log n \log d \log(1/p))$. Hence the total runtime for this part is:

$$O\left(B \log B(n + d \log n \log(1/p)) + nB \log n \log d \log(1/p) + \|\widehat{\chi}\|_0 \cdot n(\log B + \log n \log d \log(1/p))\right)$$

By Lemma 6, time to run $\textsc{RecoverFrequency}$ on all the buckets is $O(Bd \log n \log d \log(1/p))$, hence the total runtime of Algorithm 3 is,

$$O\left(B \log B \left(n + d \log n \log \frac{1}{p}\right) + nB \log n \log d \log \frac{1}{p} + \|\widehat{\chi}\|_0 \cdot n \left(\log B + \log n \log d \log \frac{1}{p}\right)\right)$$

**Sample complexity:** Samples are only consumed for computing the hashings $\widehat{u}_\sigma^a$ for all $a \in \cup_{s \in [R]} A_s$. Hence, the total sample complexity is $O(Bd \log n \log \frac{1}{p})$. $\qquad\square$

**Lemma 14.** *For any parameter $p > 0$, all integers $k$, $d$, and $b \geq \log_2(k/p)$, every signals $x, \widehat{\chi} \in \mathbb{R}^{2^n}$ such that $\|\widehat{x - \chi}\|_0 \leq k$ and $|\xi| \leq d$ for every $\xi \in \text{supp}(\widehat{x - \chi})$, the output of $\textsc{SHTInner}(x, \widehat{\chi}, p, b, d)$, $\widehat{\chi}'$ satisfies the following with probability $1 - 32p$,*

$$\|\widehat{x} - \widehat{\chi} - \widehat{\chi}'\|_0 \leq k/8.$$

*Proof.* The number of buckets is $B = 2^b \geq \frac{k}{p}$, hence, for every $f \in \text{supp}(\widehat{x} - \widehat{\chi})$ the probability of the collision event $E_{coll}(f)$ is at most $k/B \leq p$. Hence, from Lemma 13 along with a union bound it follows that, for every $f \in \text{supp}(\widehat{x - \chi})$,

$$\Pr[\widehat{\chi}'_f = \widehat{(x-\chi)}_f] \geq 1 - 2p.$$

Therefore by Markov's inequality it follows that with probability $1 - 32p$,

$$\left|\left\{f \in \operatorname{supp}(\widehat{x-\chi}) : \widehat{\chi}'_f \neq (\widehat{x-\chi})_f\right\}\right| \leq k/16,$$

It also follows from Lemma 13 that $\|\widehat{\chi}'\|_0 \leq k$, hence, $\|\widehat{x} - \widehat{\chi} - \widehat{\chi}'\|_0 \leq k/8$. $\hfill\square$

Our sparse Hadamard transform algorithm iteratively calls the primitive SHTINNER to reduces the sparsity of the residual signal by a constant factor hence it terminates in $O(\log k)$ iteration. See Algorithm 4.

---

**Algorithm 4** EXACTSHT

---

**input**: signal $x \in \mathbb{R}^{2^n}$, failure probability $q$, sparsity $k$, integer $d$.
**output**: estimate $\widehat{\chi} \in \mathbb{R}^{2^n}$.

1: $p^{(1)} \leftarrow q/32$.
2: $b^{(1)} \leftarrow \lceil \log_2 \frac{64k}{q} \rceil$.
3: $T \leftarrow \lceil \log_8 k \rceil$.
4: $w^{(0)} \leftarrow \{0\}^{2^n}$.
5: **for** $r = 1$ to $T$ **do**
6:     $\widetilde{\chi} \leftarrow \text{SHTINNER}(x, w^{(r-1)}, p^{(r)}, b^{(r)}, d)$
7:     $w^{(r)} \leftarrow w^{(r-1)} + \widetilde{\chi}$.
8:     $p^{(r+1)} \leftarrow p^{(r)}/2$.
9:     $b^{(r+1)} \leftarrow b^{(r)} - 2$.
10: $\widehat{\chi} \leftarrow w^{(T)}$.
11: **return** $\widehat{\chi}$.

---

**Proof of Theorem 2:** The proof is by induction. We denote by $\mathcal{E}_r$ the event corresponding to $\|\widehat{x} - w^{(r)}\|_0 \leq \frac{k}{8^r}$. The inductive hypothesis is that,

$$\Pr[\mathcal{E}_r | \mathcal{E}_{r-1}] \geq 1 - 16p^{(r)}.$$

Conditioned on $\mathcal{E}_{r-1}$ we have that $\|\widehat{x} - w^{(r-1)}\|_0 \leq \frac{k}{8^{r-1}}$. The number of buckets in iteration $r$ of the algorithm is $B^{(r)} = 2^{b^r} \geq \frac{64k}{4^{r-1} \cdot q}$. Hence, it follows from Lemma 14, that with probability $1 - 32p^{(r)}$,

$$\|\widehat{x} - w^{(r)}\|_0 \leq \frac{k}{8^r}.$$

This proves the inductive step.

**Runtime and Sample complexity:** In iteration $r \in [\lceil \log_8 k \rceil]$, the size of the bucket $B^{(r)} = 2^{b^{(r)}} = \frac{64k}{q \cdot 4^r}$ and the error probability $p^{(r)} = \frac{q}{32 \cdot 2^r}$. Moreover at most $\sum_r B^{(r)}$ elements are added to $\widehat{\chi}$, hence we can assume that $\|\widehat{\chi}\|_0 \leq \frac{128k}{q}$. From Lemma 13 it follows that the runtime at iteration $r$ is:

$$O\left(\frac{k}{4^r} \log \frac{k}{4^r} (n + d \log n \log 2^r) + \frac{k}{4^r} n \log n \log d \log 2^r + kn \left(\log \frac{k}{4^r} + \log n \log d \log 2^r\right)\right).$$

Summing over all $r = 1, \ldots, \lceil \log_8 k \rceil$, the total runtime is $O\left(kn \log^2 k \log n \log d\right)$.

The sample complexity of iteration $r$ is $O\left(\frac{kd}{2^r} \log n \log 2^r\right)$ hence the total sample complexity is dominated by the sample complexity of the first iteration which is equal to $O\left(kd \log n\right)$. $\hfill\square$

# 4 Robust recovery

In this section we present an algorithm to solve the robust set function learning, Problem (2). We also analyze our algorithm and prove Theorem 2. We show that our robust recovery algorithm achieves the well studied $\ell_2/\ell_2$ sparse recovery guarantee without making any assumptions on the input signal. Any general signal can be decomposed into two parts, *head* and *tail* which are defined in Definition 1. The *head* is basically the top $k$ coefficients of $\widehat{x}_f$ such that their corresponding frequencies, $f$, have Hamming weight at most $d$.

   We use hashing techniques similar to our exact recovery algorithm to solve the robust Problem (2). We can achieve the $\ell_2/\ell_2$ guarantee for general signals using $O(kd \log_2 n \log_2(d \log_2 n))$ samples which heavily relies on our novel and sample optimal primitive for recovery of low Hamming weight frequencies presented in Section 3.1.

   Note that the sample complexity of our robust algorithm in comparison to the known lower bound for this problem is only off by a $\log(d \log n)$ [PW11]. The reason for this extra factor is that when we hash a general and non-sparse signal $\widehat{x}$ into $O(k)$ buckets, the noise in each bucket might add up constructively and dominate the *head* of signal, $\widehat{x}_{head}$. Therefore we need to repeat the hashing a number of times and use median trick to get and accurate estimate to the *head* elements up to noise level.

## 4.1 Preliminaries

We consider general signals which are not necessarily sparse nor their support contains low order frequencies only. We first need to define the notion of *approximate support* for a signal. We repeat Definition 1 here.

**Definition 15** (Head and Tail norm). For all integers $n$, $d$, and $k$ we define the *head* of $\widehat{x} : \mathbb{F}_2^n \to \mathbb{R}$ as,

$$\widehat{x}_{head} := \arg \min_{\substack{y:\mathbb{F}_2^n \to \mathbb{R} \\ \|y\|_0 \leq k \\ |j| \leq d \text{ for all } j \in \mathrm{supp}(y)}} \|\widehat{x} - y\|_2.$$

The *tail norm* of $\widehat{x}$ is defined as, $\mathrm{Err}(\widehat{x}, k, d) := \|\widehat{x} - \widehat{x}_{head}\|_2^2$.

   For ease of notation, we state the next definitions and claims in terms of the residual signal $\widehat{x}' \triangleq \widehat{x} - \widehat{\chi}$ since we will be working with this signal in all iterations.

**Definition 16** (Covered frequencies). The average tail norm per bucket is denoted by $\rho$,

$$\rho := \frac{\mathrm{Err}(\widehat{x}', k, d)}{B}.$$

The set of covered frequencies is defined as,

$$S_\alpha := \{f \in \mathbb{F}_2^n : |f| \leq d \text{ and } |\widehat{x}'_f|^2 \geq \alpha \cdot \rho\}$$

for some $\alpha > 1$.

   $S_\alpha$ contains those frequencies that are recoverable by our algorithm. Intuitively, it contains all the frequencies that have large Fourier coefficients compared to average noise in each bucket. We show how recovering the frequencies in $S_\alpha$ provides a good $\ell_2/\ell_2$ approximation.

It follows from the definition of the set of covered frequencies $S_\alpha$ that if we let $\widehat{x}'_{S_\alpha}$ be the residual signal $\widehat{x}'$ restricted to set $S_\alpha$ then the following holds,

$$\begin{aligned}
\|\widehat{x}' - \widehat{x}'_{S_\alpha}\|_2^2 &= \|(\widehat{x}'_{head} + \widehat{x}'_{tail}) - (\widehat{x}'_{head \cap S_\alpha} + \widehat{x}'_{tail \cap S_\alpha})\|_2^2 \\
&= \|\widehat{x}'_{head} - \widehat{x}'_{head \cap S_\alpha}\|_2^2 + \|\widehat{x}'_{tail} - \widehat{x}'_{tail \cap S_\alpha}\|_2^2 \\
&\leq |head \setminus S_\alpha| \cdot (\alpha \rho) + \|\widehat{x}'_{tail}\|_2^2 \\
&\leq \left(1 + \alpha \cdot \frac{k}{B}\right) \cdot \mathrm{Err}(\widehat{x}', k, d).
\end{aligned} \tag{5}$$

It also follows that,

$$|S_\alpha| \leq \frac{B}{\alpha} + k. \tag{6}$$

The reason is that out of the $k$ elements in the support of $\widehat{x}'_{head}$, $S_\alpha$ can contains all of them and out of the components in the tail it can not contain more than $\frac{B}{\alpha}$ elements since this would imply the energy present in these components is at least $\frac{B}{\alpha} \cdot (\alpha \rho) = \mathrm{Err}(\widehat{x}', k, d)$ which is a contradiction. We provide an algorithm that recovers $S_\alpha$ with constant probability. We face a trade-off when picking the value of $\alpha$. If $\alpha$ is small, we are guaranteed a better approximation however the number of elements of $S_\alpha$ increase meaning extra runtime.

**Definition 17** (Collision and Large noise events). For any frequency $f \in \mathbb{F}_2^n$ and every residual signal $\widehat{x}' \in \mathbb{R}^{\mathbb{F}_2^n}$ we define two types of bad events, $E_{noise}(f)$ and $E_{coll}(f)$ corresponding to the hash function $h(f) = \sigma^\top f$,

1. Large noise, $E_{noise}(f)$: holds iff $\|\widehat{x}'_{h^{-1}(h(f)) \setminus S_\alpha}\|_2^2 \geq \beta \cdot \rho$ for some $\beta > 1$.

2. Collision, $E_{coll}(f)$: holds iff $h(f) \in h(S_\alpha \setminus \{f\})$.

If the large noise event happen, the noise might dominate the frequency $f \in S_\alpha$. The collision event happens if at least two elements of $S_\alpha$ are hashed to the same bucket.

**Claim 18** (Probability of collision and large noise). *For every $f \in \mathbb{F}_2^n$ the collision probability (see Definition 17) is upper bounded as follows,*

$$\Pr[E_{coll}(f)] \leq \frac{1}{\alpha} + \frac{k}{B}.$$

*Also the probability of large noise (see Definition 17) is bounded as follows,*

$$\Pr[E_{noise}(f)] \leq \frac{1 + \alpha(k/B)}{\beta}.$$

*The above probabilities are over the randomness of matrix $\sigma$ which is used for hashing.*

*Proof.* The probability of collision is equal to the probability that at least one of the elements of $S_\alpha \setminus \{f\}$ get hashed into the bucket $h(f)$. Hence, it follows from Claim 9 along with (6) that,

$$\Pr[E_{coll}(f)] \leq \frac{|S_\alpha|}{B} \leq \frac{1}{\alpha} + \frac{k}{B}.$$

To calculate the probability of large noise, first note that (5) implies the following for the expected noise energy per bucket (the randomness is over the hashing),

$$
\begin{aligned}
\mathbb{E}\left[\|\widehat{x}'_{h^{-1}(h(f))\setminus S_\alpha}\|_2^2\right] &= \sum_{j\in\mathbb{F}_2^n\setminus S_\alpha} |\widehat{x}'_j|^2 \cdot \Pr[j\in h^{-1}(h(f))] \\
&= \frac{1}{B}\cdot\|\widehat{x}' - \widehat{x}'_{S_\alpha}\|_2^2 \\
&\leq \left(1 + \alpha\cdot\frac{k}{B}\right)\cdot\frac{\mathrm{Err}(\widehat{x}', k, d)}{B} \\
&= \left(1 + \alpha\cdot\frac{k}{B}\right)\cdot\rho
\end{aligned}
$$

Hence, by Markov's inequality we have,

$$
\Pr\left[\|\widehat{x}'_{h^{-1}(h(f))\setminus S_\alpha}\|_2^2 \geq \beta\cdot\rho\right] \leq \frac{1 + \alpha\cdot(k/B)}{\beta}.
$$

$\square$

## 4.2 Location

In this section we design a primitive to locate any covered frequency $f\in S_\alpha$ with constant probability (Algorithm 5). More precisely, for any covered frequency $f\in S_\alpha$ if neither of the bad events $E_{coll}(f)$ and $E_{noise}(f)$ occur then $f$ can be recovered with constant probability. Moreover, the events $E_{coll}(f)$ and $E_{noise}(f)$ also occur with constant probability by Claim 18.

---

**Algorithm 5** LOCATE

---

**input**: signal $x \in \mathbb{R}^{2^n}$, signal $\widehat{\chi} \in \mathbb{R}^{2^n}$, failure probability $p$, integer $b$, integer $d$.

**output**: list of covered frequencies $L$.

1: Let $\{v^l\}_{l=0}^{\lceil \log_2 n \rceil} \subset \mathbb{F}_2^n$ be the binary search vectors on $n$ elements (Definition 4).
2: $D \leftarrow$ smallest power of two integer such that $D \geq 128d$.
3: $R \leftarrow \lceil 6 \log_2(1/p) \rceil$.
4: Let $A$ be a set of $\Theta(\log(D \log n))$ iid uniform random vectors in $\mathbb{F}_2^n$.
5: For every $r \in \{0, 1, \cdots, \log_4 D\}$ and every $s \in [R]$ let $h_r^s : [n] \to [D/2^r]$ be an independent copy of a pairwise independent hash function.
6: For every $r \in \{0, 1, \cdots, \log_4 D\}$, every $s \in [R]$, and every $j \in [D/2^r]$ let $w_{r,s}^j \in \mathbb{F}_2^n$ be the binary indicator vector of the set $h_r^s(j)^{-1}$.
7: For every $a \in A$, every $r \in \{0, 1, \cdots, \log_4 D\}$, every $s \in [R]$, every $l \in \{0, 1, \cdots, \lceil \log_2 n \rceil\}$, and every $j \in [D/2^r]$, add $a + w_{r,s}^j \cdot v^l$ to set $A_s'$.
8: Let $\sigma \in \mathbb{F}_2^{n \times b}$ be a random matrix. Each entry is independent and uniform on $\mathbb{F}_2$.
9: For every $a \in A \cup \left( \cup_{s \in [R]} A_s' \right)$ compute $\widehat{u}_\sigma^a = \text{HASH2BINS}(x, \widehat{\chi}, b, \sigma, a)$.
10: $L \leftarrow \emptyset$.
11: **for** $j = 1$ to $B$ **do**
12:      $S \leftarrow$ Empty multi-set.
13:      **for** $s \in [R]$ **do**
14:          **for** every $r \in \{0, 1, \cdots, \log_4 D\}$, every $i \in [D/2^r]$, and every $l \in \{0, 1, \cdots, \lceil \log_2 n \rceil\}$ **do**
15:              $C \leftarrow 0$.
16:              **for** every $a \in A$ **do**
17:                  **if** $\widehat{u}_\sigma^a(j)$ and $\widehat{u}_\sigma^{a+w_{r,s}^i \cdot v^l}(j)$ have same sign **then**
18:                      $C \leftarrow C + 1$.
19:              **if** $C \geq \frac{|A|}{2}$ **then**
20:                  $\phi_r^l(i) \leftarrow 0$.
21:              **else**
22:                  $\phi_r^l(i) \leftarrow 1$.
23:          $\tilde{f} \leftarrow \text{RECOVERFREQUENCY} \left( D, \{h_r^s\}_{r=0}^{\log_2 D}, \left\{ \{\phi_r^l\}_{r=0}^{\log_4 D} \right\}_{l=0}^{\lceil \log_2 n \rceil} \right)$.
24:          Append $\tilde{f}$ to multi-set $S$.
25:      $f^* \leftarrow \text{majority}(S)$.
26:      $L \leftarrow L \cup \{f^*\}$.
27: **return** $L$.

---

**Lemma 19.** *For any positive integers $n, b, d$, every signals $x, \widehat{\chi} : \mathbb{F}_2^n \to \mathbb{R}$, every $p > 0$ and every covered frequency $f \in S_\alpha$, assuming that neither $E_{coll}(f)$ nor $E_{noise}(f)$ hold, if $\alpha \geq 10\beta$ then the procedure* LOCATE *(Algorithm 5) returns a list $L$ of size $|L| \leq B$ such that,*

$$\Pr[f \in L] \geq 1 - p.$$

*Moreover the runtime of this procedure is*

$$O \left( nB \log B + (d \log B + n)B \log n \log(d \log n) \log d \log \frac{1}{p} + n\|\widehat{\chi}\|_0 (\log B + \log n \log(d \log n) \log d \log \frac{1}{p}) \right)$$

*and the number of accesses to the signal $x$ is $O \left( Bd \log n \log(d \log n) \log(1/p) \right)$.*

*Proof.* Let $j$ be the bucket that $f$ is hashed into, $j = h(f) = \sigma^\top f$. By the assumption of the lemma, $E_{coll}(f)$ doesn't hold and hence $j \notin h(S_\alpha \setminus \{f\})$. Also it is assumed that $E_{noise}(f)$ doesn't hold, hence,

$$\|\widehat{x}'_{h^{-1}(j)\setminus\{f\}}\|_2^2 < \beta\rho.$$

Therefore, for a uniformly random $a \in \mathbb{F}_2^n$ we have,

$$\mathbb{E}_a\left[\left|\widehat{u}_\sigma^a(j) - \widehat{x}'_f \cdot (-1)^{\langle a,f\rangle}\right|^2\right] = \mathbb{E}_a\left[\left|\sum_{f' \in h^{-1}(j)\setminus\{f\}} \widehat{x}'_{f'} \cdot (-1)^{\langle a,f'\rangle}\right|^2\right]$$

$$= \sum_{f' \in h^{-1}(j)\setminus\{f\}} |\widehat{x}'_{f'}|^2$$

$$< \beta\rho.$$

By Markov's inequality,

$$\Pr\left[\left|\widehat{u}_\sigma^a - \widehat{x}'_f \cdot (-1)^{\langle a,f\rangle}\right|^2 < 10\beta\rho\right] \geq 9/10.$$

By the assumption $\alpha \geq 10\beta$, the above implies that $\widehat{u}^a$ and $\widehat{x}'_f \cdot (-1)^{\langle a,f\rangle}$ have the same sign with probability $9/10$. Now, fix one $s \in [R]$. Similar to the above argument, for every $r \in \{0, 1, \cdots, \log_4 D\}$, every $i \in [D/2^r]$, and every $l \in \{0, 1, \cdots, \lceil \log_2 n \rceil\}$, $\widehat{u}^{a+w_{r,s}^i \cdot v^l}$ and $\widehat{x}'_f \cdot (-1)^{\langle a+w_{r,s}^i \cdot v^l, f\rangle}$ have the same sign with probability $9/10$. Therefore for every $a \in A$, every $r \in \{0, 1, \cdots, \log_4 D\}$, every $i \in [D/2^r]$, and every $l \in \{0, 1, \cdots, \lceil \log_2 n \rceil\}$, line 17 of Algorithm 5 correctly determines the inner product $\langle w_{r,s}^i \cdot v^l, f\rangle$ with probability $8/10$. Then Algorithm 5 uses the median trick to boost the success probability. The failure probability after taking the median over all elements of $A$ is,

$$\binom{|A|}{|A|/2} \cdot (2/10)^{|A|/2} \leq (\frac{4}{5})^{|A|/2} \leq O\left(\frac{1}{D \log n}\right).$$

By a union bound over all $r$, $l$, and $i$, the algorithm can determine the inner products $\langle w_{r,s}^i \cdot v^l, f\rangle$ simultaneously for all $r$, $l$, and $i$ with probability $1 - 1/16$. By Lemma 6, the procedure RECOVERFREQUENCY$\left(D, \{h_r^s\}_{r=0}^{\log_2 D}, \left\{\{\phi_r^l\}_{r=0}^{\log_4 D}\right\}_{l=0}^{\lceil \log_2 n \rceil}\right)$ outputs $f$ correctly with probability $1 - 1/8$. Hence, by union bound, the frequency $f$ gets recovered with probability $1 - 3/16$ in each iteration of the for loop over $s \in [R]$. The algorithm repeats this independently for every $s \in [R]$ and then takes a majority vote over all the outputted frequencies by RECOVERFREQUENCY. Frequency $f^*$ in line 25 of Algorithm 5 is the frequency which appears the most in the output of RECOVERFREQUENCY. The probability of failing to recover $f$ is the following,

$$\binom{R}{R/2} \cdot (3/16)^{R/2} \leq (3/4)^{R/2} \leq p.$$

**Runtime:** Computing the hashing $\widehat{u}_\sigma^a$ with all the different shift parameters $a \in A \cup (\cup_{s\in[R]} A'_s)$ dominates the runtime of this procedure. For a fixed $a \in \mathbb{F}_2^n$, in order to compute $x_{\sigma t+a}$ we need $B$ time samples, one for each $t \in \mathbb{F}_2^b$. The computation of the indices $\sigma t$ and all $t \in \mathbb{F}_2^b$ is upper bounded by $O(nB \log B)$ operations. Given that we have computed $\sigma t$ for all $t \in \mathbb{F}_2^b$ and stored it in memory, for a fixed $a'$ computing $\sigma t + a'$ for all $t$ takes $O(Bn)$ operation. Note that vectors $a \in \cup_{s\in[R]} A_s$ can be written as $a = a' + w_{r,s}^j \cdot v^l$ for some $r, s, j, l$ and some $a' \in A$. We can compute $\sigma t + a'$ for

all $t \in \mathbb{F}_2^b$ and all $a' \in A$ and stored it in memory in time $O(nB(\log B + \log(d \log n)))$. Note that for every $r, s, j$ the vector $w_{r,s}^j$ in line 6 of Algorithm 5 has only $2^r n/D$ non-zero entries. Therefore, given that $\sigma t + a'$ is computed and stored in memory for all $t \in \mathbb{F}_2^b$ and all $a' \in A$, we can compute $\sigma t + a$ for all $t \in \mathbb{F}_2^b$ and $a \in A \cup (\cup_{s \in [R]} A_s)$, in time $O(Bn \log n \log(d \log n) \log d \log \frac{1}{p})$. Hence, the total complexity of forming the reduced signals $x_{\sigma t + a}$ for all $a \in A \cup (\cup_{s \in [R]} A_s)$ is $O(nB(\log B + \log n \log(d \log n) \log d \log \frac{1}{p}))$. The computational complexity of a fast Walsh Hadamard transform on $x_{\sigma t + a}$ is equal to $B \log_2 B$. Hence, the computational complexity of computing the hashings $\widehat{u}_\sigma^a$ is $O(B \log B(d \log n \log(d \log n) \log(1/p) + n) + nB \log n \log(d \log n) \log d \log(1/p))$.

We also need to subtract off the current estimate $\chi$. For each frequency $f \in \text{supp}(\widehat{\chi})$ we compute $j = \sigma^\top f$. This takes $O(\|\widehat{\chi}\|_0 n \log B)$ time in total. Next for each frequency $f \in \text{supp}(\widehat{\chi})$ and each $a \in A \cup (\cup_{s \in [R]} A_s)$ the inner product $\langle f, a \rangle$ needs to be computed. This takes total time of $O(\|\widehat{\chi}\|_0 \cdot n \log n \log(d \log n) \log d \log(1/p))$. Hence the total runtime for this part is:

$$O\left(nB \log B + (d \log B + n)B \log n \log(d \log n) \log d \log \frac{1}{p} + n\|\widehat{\chi}\|_0(\log B + \log n \log(d \log n) \log d \log \frac{1}{p})\right)$$

**Sample complexity:** The sample complexity is the number of samples taken to form the hashings. The algorithm hashes the signal with $\left|A \cup \left(\cup_{s \in [R]} A'_s\right)\right|$ different shift parameters each of which requiring $B$ samples hence the total sample complexity is $O(Bd \log n \log(d \log n) \log \frac{1}{p})$. $\quad\square$

**Lemma 20.** *For every $f \in S_\alpha$, the output $L$ of the procedure* LOCATE *satisfies,*

$$\Pr[f \in L] \geq 1 - \left(p + \frac{1}{\alpha} + \frac{k}{B} + \frac{1 + \alpha(k/B)}{\beta}\right).$$

*Proof.* Note that for every $f \in S_\alpha$, by Claim 18,

$$\Pr\left[E_{coll}(f) \text{ or } E_{noise}(f)\right] \leq \frac{1}{\alpha} + \frac{k}{B} + \frac{1 + \alpha(k/B)}{\beta}.$$

Therefore the preconditions of Lemma 19 hold with probability $1 - \left(\frac{1}{\alpha} + \frac{k}{B} + \frac{1+\alpha(k/B)}{\beta}\right)$. By Lemma 19 given that its preconditions hold we have,

$$\Pr[f \in L \mid E_{coll} \text{ and } E_{noise} \text{ not holding}] \geq 1 - p.$$

Hence the lemma follows by a union bound. $\quad\square$

**Lemma 21.**
$$\mathbb{E}\left[\|\widehat{x}'_{S_\alpha \setminus L}\|_2^2\right] \leq \left(p + \frac{1}{\alpha} + \frac{k}{B} + \frac{1 + \alpha(k/B)}{\beta}\right) \cdot \|\widehat{x}'\|_2^2.$$

*Proof.* By Lemma 20 we have,

$$\mathbb{E}\left[\|\widehat{x}'_{S_\alpha \setminus L}\|_2^2\right] = \sum_{f \in S_\alpha} |\widehat{x}'_f|^2 \cdot \Pr[f \notin L]$$

$$\leq \sum_{f \in S_\alpha} |\widehat{x}'_f|^2 \cdot \left(2p + \frac{1}{\alpha} + \frac{k}{B} + \frac{1 + \alpha(k/B)}{\beta}\right)$$

$$\leq \left(2p + \frac{1}{\alpha} + \frac{k}{B} + \frac{1 + \alpha(k/B)}{\beta}\right) \cdot \|\widehat{x}'\|_2^2.$$

$\quad\square$

If we let $\beta = \frac{10}{\delta q}$ and $\alpha = \beta$ and $B = 10\alpha k$ and run the procedure LOCATE with failure probability $p = \delta q/10$, then by Markov's inequality the following holds,

$$\Pr\left[\|\widehat{x}'_{S_\alpha \setminus L}\|_2^2 \leq \delta \cdot \|\widehat{x}'\|_2^2\right] \geq 1 - q.$$

## 4.3 Estimation

In this section we use the hashing technique to estimate the values of the signal $\widehat{x}$ at frequencies $f \in L$ for some set of locations $L \subset \mathbb{F}_2^n$. This is done in Algorithm 6.

---
**Algorithm 6** ESTIMATE
---
**input**: signal $x \in \mathbb{R}^{2^n}$, signal $\widehat{\chi} \in \mathbb{R}^{2^n}$, failure probability $p$, integer $b$, list of frequencies $L$, integer $k$, parameter $\gamma$.
**output**: estimated signal $\widehat{\chi}'$.

1: $B \leftarrow 2^b$.
2: $T \leftarrow O(\log \frac{B}{p\gamma k})$.
3: **for** $r = 1$ to $T$ **do**
4:      Let $a_r$ be a uniformly random vector in $\mathbb{F}_2^n$.
5:      Let $\sigma_r \in \mathbb{F}_2^{n \times b}$ be a random matrix. Each entry is independent and uniform on $\mathbb{F}_2$.
6:      Compute $\widehat{u}_{\sigma_r}^{a_r} = \text{HASH2BINS}(x, \widehat{\chi}, b, \sigma_r, a_r)$.
7: $\widehat{w} \leftarrow \{0\}^{2^n}$.
8: **for** $f \in L$ **do**
9:      $\widehat{w}_f \leftarrow \text{median}_{r \in [T]}\left(\widehat{u}_{\sigma_r}^{a_r}(\sigma_r^\top f) \cdot (-1)^{\langle f, a_r \rangle}\right)$.
10: $J \leftarrow \arg\max_{J:|J|=2k} \|\widehat{w}_J\|_2^2$
11: $\widehat{\chi}' \leftarrow \widehat{w}_J$.
12: **return** $\widehat{\chi}'$.

---

**Lemma 22** (Estimation guarantee for a single frequency). *For any $L \subset \mathbb{F}_2^n$ with size $|L| \leq B$, any $\gamma > 0$, and any $f \in L$, if $\beta \geq 40$ and $\alpha \geq 4\beta$ and $B \geq 10\alpha k$ then*

1. $\Pr\left[\left|\widehat{x}'_f - \widehat{w}_f\right|^2 \geq 6\beta\rho\right] \leq p\gamma\frac{k}{B}$, *where $\widehat{x}' = \widehat{x} - \widehat{\chi}$ is the input signal to the estimation procedure.*

2. $\Pr\left[Err(\widehat{x}'_L - \widehat{\chi}', \gamma k) \leq Err(\widehat{x}'_L, k) + 24\beta k\rho\right] \geq 1 - p.$

*Moreover the runtime of this procedure is $O\left(Bn\log(B/p\gamma k)\log B + \|\widehat{\chi}\|_0 \cdot n\log(B/p\gamma k)\log B\right)$ and the number of accesses to the signal $x$ is $O\left(B \cdot \log(B/p\gamma k)\right)$.*

*Proof.* For every $f \in L$ and every $r$, let $j = \sigma_r^\top f$ be the bucket that $f$ is hashed into by the $r^{th}$ hash function. Note that for every $r$, with probability $1 - \left(1/\alpha + k/B + \frac{1+\alpha(k/B)}{\beta}\right)$, neither of the events $E_{coll}(f)$ and $E_{noise}(f)$ hold. Conditioning on $E_{coll}(f)$ and $E_{noise}(f)$ not holding, we have,

$$\mathbb{E}_{a_r}\left[\left|\widehat{x}'_f - \widehat{u}_{\sigma_r}^{a_r} \cdot (-1)^{\langle f, a_r \rangle}\right|^2\right] = \mathbb{E}\left[\left|\sum_{f' \in h^{-1}(j)\setminus\{f\}} \widehat{x}'_{f'} \cdot (-1)^{\langle f, a_r \rangle}\right|^2\right]$$

$$\leq \sum_{f' \in h^{-1}(j)\setminus\{f\}} |\widehat{x}'_{f'}|^2$$

$$\leq \beta\rho.$$

Therefore by Markov's inequality and a union bound,

$$Pr\left[\left|\widehat{x}'_f - \widehat{u}^{a_r}_{\sigma_r} \cdot (-1)^{\langle f, a_r \rangle}\right|^2 \geq 6\beta\rho\right] \leq 1/6 + 1/\alpha + k/B + \frac{1 + \alpha(k/B)}{\beta} \leq 1/5.$$

Hence when we take the median of $\widehat{u}^{a_r} \cdot (-1)^{\langle f, a_r \rangle}$ for all $r$ the error probability goes down as follows,

$$\Pr\left[\left|\widehat{x}'_f - \widehat{w}_f\right|^2 \geq 6\beta\rho\right] \leq \binom{T}{T/2} \cdot (\frac{1}{5})^{T/2} \leq (\frac{4}{5})^{T/2} \leq p\gamma k/(2B).$$

This proves the first claim of the lemma.

Let $U = \left\{f \in L : \left|\widehat{x}'_f - \widehat{w}_f\right|^2 \geq 6\beta\rho\right\}$. It follows from the first claim of the lemma along with Markov's inequality that $|U| \leq |L| \cdot \gamma k/B \leq \gamma k$ with probability $1 - p/2$. Conditioning on this happening we have,

$$\|(\widehat{x}' - \widehat{w})_{L \setminus U}\|^2_\infty \leq 6\beta\rho.$$

Let $T$ denote the top $k$ coordinates of $\widehat{w}_{L \setminus U}$. It follows from the above that,

$$\|\widehat{x}'_{L \setminus U} - \widehat{w}_T\|^2_2 \leq \mathrm{Err}(\widehat{x}'_{L \setminus U}, k) + (3k) \cdot (6\beta\rho)$$
$$\leq \mathrm{Err}(\widehat{x}'_L, k) + 18\beta k\rho. \qquad (7)$$

Because $J$ is the top $2k \geq (1 + \gamma)k$ coordinates of $\widehat{w}$, $T \subseteq J \setminus U$. Let $J' = J \setminus (T \cup U)$, so $|J'| \leq k$. Therefore,

$$\mathrm{Err}(\widehat{x}_L - \widehat{\chi}', \gamma k) \leq \|\widehat{x}'_{L \setminus U} - \widehat{\chi}'_{J \setminus U}\|^2_2$$
$$= \|\widehat{x}_{L \setminus (U \cup J')} - \widehat{\chi}'_T\|^2_2 + \|(\widehat{x} - \widehat{\chi}')_{J'}\|^2_2$$
$$\leq \|\widehat{x}_{L \setminus U} - \widehat{\chi}'_T\|^2_2 + |J'| \cdot \|(\widehat{x} - \widehat{\chi}')_{J'}\|^2_\infty$$
$$\leq \mathrm{Err}(\widehat{x}_L, k, d) + 18\beta k\rho + 6\beta k\rho$$
$$= \mathrm{Err}(\widehat{x}_L, k, d) + 24\beta k\rho.$$

Third inequality above follows from (7).

**Runtime:** The runtime is dominated by the time to compute the hashings $\widehat{u}^{a_r}_{\sigma_r}$ for every $r \in [T]$. By Claim 10, this computation takes a total of $O(Bn \log_2(B/kp\gamma) \log B + \|\widehat{\chi}\|_0 \cdot n \log_2(B/kp\gamma) \log B)$ operations.

**Sample complexity:** The sample complexity is the number of samples taken to form the hashings $u^{a_r}_{\sigma_r}$. The algorithm uses $O(\log(B/kp\gamma))$ hashes each of which requiring $B$ samples and hence the total sample complexity is $O(B \log(B/kp\gamma))$. $\qquad \square$

## 4.4 Reduce SNR

In this section we put the primitives LOCATE and ESTIMATE together to design a procedure which reduces the norm of the head of the signal $\|\widehat{x}_{head}\|_2$ by a large constant factor while the norm of tail $\mathrm{Err}(\widehat{x}, k)$ only mildly increases hence the signal to noise ratio decreases by a constant factor. Algorithm 7 does this task. We also show that the residual $\widehat{x} - \widehat{\chi}'$ after running this procedure is sparser than the original signal $\widehat{x} - \widehat{\chi}$.

**Algorithm 7** REDUCESNR

---

**input**: signal $x \in \mathbb{R}^{2^n}$, signal $\widehat{\chi} \in \mathbb{R}^{2^n}$, parameter $\delta > 0$, failure probability $q$, sparsity $k$, integer $d$, parameter $\gamma$.

**output**: refined estimate $\widehat{\chi}'$.

1: $p \leftarrow q\gamma\delta/10$.
2: $b \leftarrow \left\lceil \log_2(\frac{10k}{p\delta}) \right\rceil$.
3: $L \leftarrow \text{LOCATE}(x, \widehat{\chi}, p, b, d)$.
4: $L' \leftarrow \{f \in L : |f| \leq d\}$.
5: $\widetilde{\chi} \leftarrow \text{ESTIMATE}(x, \widehat{\chi}, p, b, L', k, d, \gamma)$.
6: $\widehat{\chi}' \leftarrow \widehat{\chi} + \widetilde{\chi}$.
7: **return** $\widehat{\chi}'$.

---

**Lemma 23.** *For all integers $n$, $k$, and $d$, every parameters $0 < \gamma < 1/2$, $0 < \delta < 0.1$, and $0 < q < 1/2$, every signals $x, \widehat{\chi} : \mathbb{F}_2^n \to \mathbb{R}$, the procedure* REDUCESNR *outputs a signal $\widehat{\chi}' : \mathbb{F}_2^n \to \mathbb{R}$ such that the following holds,*

$$\Pr\left[ Err(\widehat{x} - \widehat{\chi}', 2\gamma k, d) \leq (1 + 4\delta) \cdot Err(\widehat{x} - \widehat{\chi}, k, d) \right] \geq 1 - q.$$

*Moreover, the runtime of this procedure is*

$$O\left( \log \frac{1}{q\gamma\delta} \left( \frac{k}{q\gamma\delta^2} (n \log \frac{k}{q\gamma\delta} + (d \log \frac{k}{q\gamma\delta} + n) \log n \log(d \log n) \log d) + n\|\widehat{\chi}\|_0 (\log \frac{k}{q\gamma\delta} + \log n \log(d \log n) \log d) \right) \right)$$

*and the sample complexity of the procedure is, $O\left( \frac{kd}{q\gamma\delta^2} \cdot \log n \cdot \log(d \log n) \cdot \log \frac{1}{q\gamma\delta} \right)$.*

*Proof.* Let $\beta = 1/p$, $\alpha = 10\beta$, and $B = (\alpha/\delta) \cdot k = \frac{10k}{p\delta}$. Let $S_\alpha$ be the set of covered frequencies of $\widehat{x}' = \widehat{x} - \widehat{\chi}$. Let us denote the output of $\text{ESTIMATE}(x, \widehat{\chi}, p, b, L', k, \gamma)$ by $\widetilde{\chi}$. The support of $\widetilde{\chi}$ is denoted by $J = \text{supp}(\widetilde{\chi})$. Also let $L$ be the output of the procedure $\text{LOCATE}(x, \widehat{\chi}, p, b, d)$. Since by definition of covered frequencies, $|f| \leq d$ for every $f \in S_\alpha$, by Lemma 20 we have,

$$\Pr[f \in L' | f \in S_\alpha] = \Pr[f \in L | f \in S_\alpha] \geq 1 - q\gamma\delta/2.$$

The above along with $|S_\alpha| \leq (1 + 1/\delta)k$, which follows from (6), and Markov's inequality gives the following,

$$\Pr[|S_\alpha \setminus L'| \leq \gamma k] \geq 1 - 2q/3.$$

Therefore, conditioning on the above event,

$$\begin{aligned}
\text{Err}(\widehat{x}' - \widehat{x}'_{L'}, \gamma k, d) &\leq \|\widehat{x}' - \widehat{x}'_{(L' \cup S_\alpha)}\|_2^2 \\
&\leq \text{Err}(\widehat{x}' - \widehat{x}'_{(L' \cup S_\alpha)}, k, n) + k \cdot \|\widehat{x}' - \widehat{x}'_{(L' \cup S_\alpha)}\|_\infty^2 \\
&\leq \text{Err}(\widehat{x}' - \widehat{x}'_{L'}, k, n) + k \cdot \|\widehat{x}' - \widehat{x}'_{S_\alpha}\|_\infty^2 \\
&\leq \text{Err}(\widehat{x}' - \widehat{x}'_{L'}, k, d) + k \cdot \alpha\rho,
\end{aligned}$$

where the first inequality follows because $\forall f \in S_\alpha \setminus L'$, $|f| \leq d$. Note that $\text{supp}(\widetilde{\chi}) \subseteq L$, hence,

$$\begin{aligned}
\text{Err}(\widehat{x} - \widehat{\chi}', 2\gamma k, d) &= \text{Err}(\widehat{x}' - \widetilde{\chi}, 2\gamma k, d) \\
&\leq \text{Err}(\widehat{x}'_{L'} - \widetilde{\chi}, \gamma k, d) + \text{Err}(\widehat{x}' - \widehat{x}'_{L'}, \gamma k, d) \\
&\leq \text{Err}(\widehat{x}'_{L'} - \widetilde{\chi}, \gamma k, d) + \text{Err}(\widehat{x}' - \widehat{x}'_{L'}, k, d) + \alpha k\rho
\end{aligned}$$

By second part of Lemma 22, $\text{Err}(\widehat{x}'_{L'} - \widetilde{\chi}, \gamma k, d) \leq \text{Err}(\widehat{x}'_{L'}, k, d) + 24\beta k\rho$ with probability $1 - p$, therefore,

$$
\begin{aligned}
\text{Err}(\widehat{x} - \widehat{\chi}', 2\gamma k, d) &\leq \text{Err}(\widehat{x}'_{L'}, k, d) + 24\beta k\rho + \text{Err}(\widehat{x}' - \widehat{x}'_{L'}, k, d) + \alpha k\rho \\
&\leq \text{Err}(\widehat{x}'_{L'}, k, d) + \text{Err}(\widehat{x}' - \widehat{x}'_{L'}, k, d) + 34\beta k\rho \\
&\leq \text{Err}(\widehat{x}', k, d) + 34\beta k\rho \\
&\leq (1 + 4\delta)\text{Err}(\widehat{x}', k, d).
\end{aligned}
$$

By a union bound over the randomness of LOCATE and ESTIMATE primitives the above holds with probability at least $1 - q$.

**Runtime and Sample complexity:** The runtime and sample complexity follow from invoking Lemma 19 and Lemma 22 with $B = O(\frac{k}{q\delta^2\gamma})$ and $p = O(q\gamma\delta)$.

$\qquad\qquad\qquad\qquad\qquad\qquad\qquad\qquad\qquad\qquad\qquad\qquad\qquad\qquad\qquad\qquad\qquad\quad$ $\square$

## 4.5 Iterative Peeling

Here we present an iterative algorithm which reduces the SNR of the input signal as well as its sparsity by a constant factor in each iteration and hence terminates in $O(\log_2 k)$ rounds.

---

**Algorithm 8** ROBUSTSHT

---

**input**: signal $x \in \mathbb{R}^{2^n}$, parameter $\delta > 0$, failure probability $q$, sparsity $k$, integer $d$.
**output**: estimate $\widehat{\chi} \in \mathbb{R}^{2^n}$.

1: $\gamma \leftarrow 1/64$.
2: $T \leftarrow \lceil \log_{1/\gamma} k \rceil$.
3: $q^{(1)} \leftarrow q/2$.
4: $\delta^{(1)} \leftarrow \delta/20$.
5: $k^{(1)} \leftarrow k$.
6: $w^{(0)} \leftarrow \{0\}^{2^n}$.
7: **for** $r = 1$ to $T$ **do**
8: $\quad w^{(r)} \leftarrow \text{REDUCESNR}(x, w^{(r-1)}, \delta^{(r)}, q^{(r)}, k^{(r)}, d, \gamma/2)$.
9: $\quad \delta^{(r+1)} \leftarrow \delta^{(r)}/2$.
10: $\quad q^{(r+1)} \leftarrow q^{(r)}/2$.
11: $\quad k^{(r+1)} \leftarrow \gamma \cdot k^{(r)}$.
12: $\widehat{\chi} \leftarrow w^{(T)}$.
13: **return** $\widehat{\chi}$.

---

**Proof of Theorem 3:** The proof is by induction. The induction hypothesis is that for every iteration $r$ the followings hold,

$$
\Pr\left[\mathcal{E}_r | \mathcal{E}_{r-1}\right] \geq 1 - q^{(r)},
$$

where event $\mathcal{E}_r$ for every $r$ is defined as the following,

$$
\mathcal{E}_r = \begin{cases} 1) \ |\text{supp}(w^{(r)})| \leq \sum_{t=0}^{r} 2k^{(t)} \\ 2) \ \text{Err}(\widehat{x} - w^{(r)}, \gamma k^{(r)}, d) \leq \prod_{t=0}^{r}\left(1 + 3\delta^{(t)}\right) \cdot \text{Err}(\widehat{x}, k, d) \end{cases} .
$$

It follows from Lemma 23 that the inductive hypothesis holds for every $r$. Therefore by a union bound we have the following,

$$\Pr\left[\bar{\mathcal{E}}_T\right] \leq \sum_{r=1}^{T} \Pr\left[\bar{\mathcal{E}}_r | \mathcal{E}_{r-1}\right] + \Pr\left[\bar{\mathcal{E}}_0\right]$$

$$\leq \sum_{r=1}^{T} q^{(r)}$$

$$\leq q.$$

Conditioning on $\mathcal{E}_T$ happening which occurs with probability $1 - q$ the following holds,

$$|\text{supp}(\widehat{\chi})| \leq \sum_{r=1}^{T} 2k^{(r)} \leq 3k.$$

And also,

$$\text{Err}(\widehat{x} - \widehat{\chi}, 0) = \|\widehat{x} - \widehat{\chi}\|_2^2$$

$$\leq \prod_{r=1}^{T} \left(1 + 4\delta^{(r)}\right) \cdot \text{Err}(\widehat{x}, k)$$

$$\leq (1 + \delta) \cdot \text{Err}(\widehat{x}, k).$$

This concludes the first and third claims of the theorem. Second claim of the theorem follows from line 4 of Algorithm 7.

**Runtime and Sample complexity:** At iteration $r$ of the algorithm the values of the input parameters to REDUCESNR primitive are as follows, $\delta^{(r)} = O(\delta/2^r)$, $q^{(r)} = O(q/2^r)$, $k^{(r)} = O(k/64^r)$ and $\gamma = O(1)$ and also it follows from the inductive proof that $\|w^{(t)}\|_0 = O(k)$. Hence by Lemma 23, the runtime of this iteration is the following,

$$O\left(\log 4^r \left(\frac{k}{8^r \delta^2}(n \log k + (d \log k + n) \log n \log(d \log n) \log d) + nk(\log k + \log n \log(d \log n) \log d)\right)\right).$$

Therefore the total runtime is $O\left(nk \log^3 k + nk \log^2 k \log n \log(d \log n) \log d\right)$. Also the sample complexity of the $r^{th}$ iteration is $O\left(\frac{kd}{8^r \delta^2} \log n \log(d \log n) \log 4^r\right)$ which leads to a total sample complexity of $O\left(\frac{kd}{\delta^2} \log n \log(d \log n)\right)$. $\square$

# 5  Experiments

We test our EXACTSHT algorithm for graph sketching on a real world data set. We utilize the autonomous systems dataset from the SNAP data collection.[1] In order to compare our methods with [SK12] we reproduce their experimental setup. The dataset consists of 9 snapshots of an autonomous system in Oregon on 9 different dates. The goal is detect which edges are added and removed when comparing the system on two different dates. As a pre-processing step, we find the common vertices that exist on all dates and look at the induced subgraphs on these vertices. We take the symmetric differences (over the edges) of dates 7 and 9. Results for other date combinations can be found in the supplementary material. This results in a sparse graph (sparse in the number of edges). Recall that the running time of our algorithm is $O(kn \log^2 k \log n \log d)$ which reduces to $O(nk \log^2 k \log n)$ for the case of cut functions where $d = 2$.

Table 1: Sampling and computational complexity

| No. of vertices | CS method | | Our method | |
|---|---|---|---|---|
| | Runtime | Samples | Runtime | Samples |
| 70 | 1.14 | 767 | 0.85 | 6428 |
| 90 | 1.88 | 812 | 0.92 | 6490 |
| 110 | 3.00 | 850 | 0.82 | 6491 |
| 130 | 4.31 | 880 | 1.01 | 7549 |
| 150 | 5.34 | 905 | 1.16 | 7942 |
| 170 | 6.13 | 927 | 1.22 | 7942 |
| 190 | 7.36 | 947 | 1.18 | 7271 |
| 210 | 8.24 | 965 | 1.28 | 7271 |
| 230 | * | * | 1.38 | 7942 |
| 250 | * | * | 1.38 | 7271 |
| 300 | * | * | 1.66 | 8051 |
| 400 | * | * | 2.06 | 8794 |
| 500 | * | * | 2.42 | 8794 |
| 600 | * | * | 3.10 | 9646 |
| 700 | * | * | 3.35 | 9646 |
| 800 | * | * | 3.60 | 9646 |

## 5.1 Sample and time complexities as number of vertices varies

In the first experiment depicted in Figures 1b-4b we order the vertices of the graph by their degree and look at the induced subgraph on the $n$ largest vertices in terms of degree where $n$ varies. For each $n$ we pick $e = 50$ edges uniformly at random. The goal is to learn the underlying graph by observing the values of cuts. We choose parameters of our algorithm such that the probability of success is at least 0.9. The parameters tuned in our algorithm to reach this error probability are the initial number of buckets the frequencies are hashed to and the ratio at which they reduce in each iteration. We plot running times as $n$ varies. We compare our algorithm with that of [SK12] which utilizes a CS approach. We fine-tune their algorithm by the only tuneable parameter which is sampling complexity. Both algorithms are run in a way such that each sample (each observation of a cut value) takes the same time. As one can see our algorithm scales *linear* in $n$ whereas the CS approach scales *quadratically*. Our algorithm continues to work in a reasonable amount of time for vertex sizes as much as 900 in under 2 seconds. The error bars show the standard deviation of the running times.

In Table 1 we include both sampling complexities (number of observed cuts) and running times as $n$ varies. Our sampling complexity is equal to $O(k \log n)$. In practice they perform around a constant factor of 10 worse than the compressive sensing method, which are not provably optimal (see Section 1) but perform well in practice. In terms of computational cost, however, the CS approach quickly becomes intractable, taking large amounts of time on instance sizes around 200 and larger [SK12]. Asterisks in Table 1 refer to experiments that have taken too long to be feasible to run.

## 5.2 Time complexities as number of edges varies

Here we fix the number of vertices to $n = 100$ and consider the induced subgraph on these vertices. We randomly pick $e$ edges to include in the graph. We plot computational complexities. Our running time provably scales linearly in the number of edges as can be seen in Figures 1a-4a.

(a) Avg. time vs. no. edges

(b) Avg. time vs. no. vertices

Figure 1: Comparison of scaling of runtimes of our algorithm vs CS approaches

(a) Avg. time vs. no. edges

(b) Avg. time vs. no. vertices

Figure 2: Comparison of scaling of runtimes of our algorithm vs CS approaches

(a) Avg. time vs. no. edges

(b) Avg. time vs. no. vertices

Figure 3: Comparison of scaling of runtimes of our algorithm vs CS approaches

(a) Avg. time vs. no. edges

(b) Avg. time vs. no. vertices

Figure 4: Comparison of scaling of runtimes of our algorithm vs CS approaches

## Footnotes

[1]snap.stanford.edu/data/