[Reviews · NeurIPS 2019]

Reviewer 1



The main advance of this manuscript is the reduced sampling complexity and computational speed of the proposed algorithms ExactSHT and RobustSHT; with theoretical guarantees. These advances are illustrated on a small scale toy problem, see Figure on page 8. One of the deficiencies of the manuscript is the very limited quality of numerical experiments, so much as to not include some of the important aspects such as probability of recovery and the differing performance of the ExactSHT and RobustSHT. The main manuscript includes a pair of plots with a small discussion, and the supplementary material includes a modestly longer experimental section; see for instance Table 1 on page 23. The advances are important, more so than many results in compressed sensing these days as CS is a well developed topic. The results are presented clearly, but the organisation of the results isn't ideal at present as the sampling complexity isn't shown in the main manuscript. The authors are encouraged to move Table 1 of the supplementary material to the main manuscript as this is especially important; alternatively it could be presented in plot form within the current plots using a right axis on sampling complexity.

Reviewer 2



I do not feel qualified enough to review the technical aspects of this submission, but reading the paper makes me sligthly uneasy. - The very first sentence of the introduction is almost verbatim from another paper (first sentence of the abstract of [11]) - Also; - some clear overselling: "a natural and very important class to consider" (based on one paper about hyper-parameter tuning for deep nets) seems like a tenuous motivation. - specifying in all O(.)-type result the base of the logarithm seems.. strange, and does not really inspire confidence ("log_2" everywhere in big_Oh notations...) - Theorem 2 has no approximation parameter. I do not see how, given query access to a function x, one can output the *exact* values of the non-zero coefficients of \hat{x} in a sublinear number of queries (which is what "problem (1)" asks) - no comparison of the results with what is obtained in [8], which gives results on learning Fourier-sparse functions. While I may very well be wrong, the above points make me believe this submission should be rejected. UPDATE: In light of the other reviews, and the authors' response, I have updated my score to reflect better the importance of the problem solved, and the response of the authors regarding the points made above (also, I would suggest to include, in some form, part of the answer (2) in the paper itself). However, I would very much like to point out again that, regardless of the results, verbatim lifting of other papers' writing with no acknowledgment is not OK (even in the case of self-plagiarism, a point anyway moot here given the anonymity). This played a significant part in my original assessment.

Reviewer 3



The results are clever and nice. They are a bit similar to the results in "New Results for Learning Noisy Parities and Halfspaces" (Theorem 1 where in that paper they isolate large Fourier coefficients). Here you must be very careful with the hashing scheme to get the optimal bounds.

[Author Response · NeurIPS 2019]

We thank all the reviewers for their time and constructive comments.

**Reviewer 3:** We will move the experiments regarding the sampling complexity (table 1) and some discussions from
the supplementary material to the main manuscript. Indeed, due to the 8-page constraint, the table and some of the
discussions were moved to the supplementary material so that the algorithm could be fully presented in the main paper.
We also planned on moving the table and more in detailed discussion back for the final paper if accepted.

**Reviewer 4:**

**(1)** We will make sure to reword/rewrite the first sentence of our intro.

**(2)** The problem of learning Fourier sparse set functions whose Fourier support contain low Hamming weight frequencies
arises in many domains. One application of this problem is learning sparse graphs/hypergraphs from observing their
cut values [11]. Another application of this problem is learning decision trees with bounded depth [7,8]. Moreover,
hyper-parameter tuning of deep nets can be cast as a decision tree learning problem [5] and hence our results can be
used for this problem as well. From a higher level perspective, the technique of splitting a multidimensional function
into a summation of (say $k$) functions each depending on a small subset of the input variables (say $d$ variables) is a
common assumption that makes learning tractable. Our algorithm allows for the learning of set functions that have this
form with information-theoretically tight rates and optimal running times. We believe this itself is an important enough
contribution and could open up many paths for further applications.

All in all, we will make sure to rewrite the motivation part of the intro and mention all these applications more clearly.
The sentence "Therefore, the family of Fourier sparse set functions whose Fourier support only contains low order
terms is a natural and very important class to consider" Will be replaced by: "Therefore, the family of Fourier sparse set
functions whose Fourier support only contains low order term is viewed as a natural and important class of functions to
consider".

**(3)** In some places, such as the algorithm descriptions, it is important to specify the base of the $\log$'s since there is
no Big-Oh notation there. We decided to specify the base of the $\log$'s everywhere for uniformity of presentation, but
certainly agree that this is not needed in the Big-Oh notation.

**(4)** Exact recovery of sparse functions in sublinear time is indeed possible. Consider a 1-sparse function, $x(t) =$
$\widehat{x}(f) \cdot (-1)^{f^\top t}$ for all $t \in \{0,1\}^n$ and some frequency $f \in \{0,1\}^n$. Consider $x(0^n) = \widehat{x}(f)$ and $x(t) = \widehat{x}(f) \cdot (-1)^{f^\top t}$
for some $t \neq 0^n$. One can observe that $x(0^n)$ and $x(t)$ have the same absolute value and potentially different signs. If
their signs differ then $(-1)^{f^\top t}$ must be $-1$ and hence $f^\top t = 1$, and if their signs agree then $f^\top t = 0$. Therefore, by
comparing the sign of $x(0^n)$ to the sign of $x(t)$, it is possible to learn $f^\top t$. By learning $f^\top t$ for all $t \in \{e_1, e_2, \cdots e_n\}$,
where $e_i \in \{0,1\}^n$ is the $i^{th}$ standard basis vectors with $i^{th}$ entry equal to 1 and the rest of entries equal to 0, one can
learn the frequency $f$ bit by bit, using $n$ samples and runtime. We show in section 3.1 that if the frequency $f$ has a low
Hamming weight, $|f| \leq d$, then we can recover $f$ more efficiently using $O(d \log n)$ samples and runtime as opposed to
$n$. If $x$ is $k$-sparse then the idea is to hash its Fourier transform $\widehat{x}$ into $Ck$ buckets for some large constant $C$ and then
run this 1-sparse recovery primitive on each bucket. A constant fraction of the non-zero Fourier coefficients will be
isolated in their buckets and hence the 1-sparse recovery will find them successfully. Then we repeat this procedure
$O(\log k)$ times and in each round, the sparsity goes down by a constant factor. This general hashing and iterative
peeling technique have been used for sublinear time sparse FFT of 1-D signals [3]. We solve the problem of sparse
Fourier transform on the binary hypercube $\{0,1\}^n$ (Hadamard transform) and achieve the optimal sample complexity
and runtime for recovery of frequencies with bounded weight.

**(5)** The result of [8] is not able to exactly recover sparse set functions in the noiseless setting. It also has a weaker
recovery guarantee than [11] in the robust setting. Therefore, we compare our results against [11] which provides the
stronger $\ell_2/\ell_2$ approximation guarantee and supports exact recovery in the noiseless setting.

**Reviewer 5:**

We thank the reviewer for the acknowledgment of our hashing schemes. Indeed, as an important contribution, we have
removed assumptions on the randomness of the support as compared to previous work by Scheibler [10].

Our hash function (Definition 7) is in fact similar to the *projection* defined in the paper "New Results for Learning Noisy
Parities and Halfspaces". Theorem 7 of this paper proves that the problem of finding a $\theta$-heavy Fourier coefficient
can be reduced to the problem of learning parities under uniform distribution and classification noise. However, this
problem is not known to be solvable in time $\text{poly}(n)$ – in fact, this problem is closely related to the *learning with errors*,
LWE problem, whose hardness has been used as a cryptographic assumption. Hence these techniques do not yield
efficient recovery. We will add a discussion about this in the final version of our manuscript.

[Meta-Review · NeurIPS 2019]

The main contribution of this paper is an information theoretically optimal algorithm (in terms of the number of queries it makes to the function) for learning a sparse fourier approximation. This is a well-studied problem that is at the heart of many applications in property testing. The reviewers all felt that the problem is important and the techniques are clever and were enthusiastic about this paper.